# TABPFN: A TRANSFORMER THAT SOLVES SMALL TABULAR CLASSIFICATION PROBLEMS IN A SECOND

**Noah Hollmann**[*,1,2]  **Samuel Müller**[*,1]  **Katharina Eggensperger**[1]  **Frank Hutter**[1,3]
[1] University of Freiburg, [2] Charité University Medicine Berlin
[3] Bosch Center for Artificial Intelligence [*] Equal contribution.
Correspondence to `noah.hollmann@charite.de` & `muellesa@cs.uni-freiburg.de`

## ABSTRACT

We present TabPFN, a trained Transformer that can do supervised classification for small tabular datasets in *less than a second*, needs no hyperparameter tuning and is competitive with state-of-the-art classification methods. TabPFN is fully entailed in the weights of our network, which accepts training and test samples as a set-valued input and yields predictions for the entire test set in a single forward pass. TabPFN is a Prior-Data Fitted Network (PFN) and is trained offline once, to approximate Bayesian inference on synthetic datasets drawn from our prior. This prior incorporates ideas from causal reasoning: It entails a large space of structural causal models with a preference for simple structures. On the 18 datasets in the OpenML-CC18 suite that contain up to 1 000 training data points, up to 100 purely numerical features without missing values, and up to 10 classes, we show that our method clearly outperforms boosted trees and performs on par with complex state-of-the-art AutoML systems with up to 230× speedup. This increases to a 5 700× speedup when using a GPU. We also validate these results on an additional 67 small numerical datasets from OpenML. We provide all our code, the trained TabPFN, an interactive browser demo and a Colab notebook at `https://github.com/automl/TabPFN`.

## 1 INTRODUCTION

Tabular data has long been overlooked by deep learning research, despite being the most common data type in real-world machine learning (ML) applications (Chui et al., 2018). While deep learning methods excel on many ML applications, tabular data classification problems are still dominated by Gradient-Boosted Decision Trees (GBDT; Friedman, 2001), largely due to their short training time and robustness (Shwartz-Ziv and Armon, 2022).

We propose a radical change to how tabular classification is done. We do *not* fit a new model from scratch to the training portion of a new dataset. Instead, we replace this step by performing a single forward pass with a large Transformer that has been pre-trained to solve artificially generated classification tasks from a tabular dataset prior.

Our method builds on Prior-Data Fitted Networks (PFNs; Müller et al., 2022; see Section 2), which learn the training and prediction algorithm itself. PFNs approximate Bayesian inference given any prior one can sample from and approximate the posterior predictive distribution (PPD) directly. While inductive biases in NNs and GBDTs depend on them being efficient to implement (e.g., through $L_2$ regularization, dropout (Srivastava et al., 2014) or limited tree-depth), in PFNs, one can simply design a dataset-generating algorithm that encodes the desired prior. This fundamentally changes the way we can design learning algorithms.

We design a prior (see Section 4) based on Bayesian Neural Networks (BNNs; Neal 1996; Gal 2016) and Structural Causal Models (SCMs; Pearl 2009; Peters et al. 2017) to model complex feature dependencies and potential causal mechanisms underlying tabular data. Our prior also takes ideas from Occam's razor: simpler SCMs and BNNs (with fewer parameters) have a higher likelihood. Our prior is defined via parametric distributions, e.g., a log-scaled uniform distribution for the average number of nodes in data-generating SCMs. The resulting PPD implicitly models uncertainty over

all possible data-generating mechanisms, weighting them by their likelihood given the data and their prior probability. Thus, the PPD corresponds to an infinitely large ensemble of data-generating mechanisms, i.e., instantiations of SCMs and BNNs. We learn to approximate this complex PPD in a single forward-pass, requiring no cross-validation or model selection.

Our **key contribution** is to introduce the *TabPFN* (see Section 3), a single Transformer that has been pre-trained to approximate probabilistic inference for the novel prior above (described in more detail in Section 4) in a single forward pass, and has thus learned to solve novel small tabular classification tasks ($\leq 1\,000$ training examples, $\leq 100$ purely numerical features without missing values and $\leq 10$ classes) in *less than a second* yielding state-of-the-art performance.

To substantiate this claim, we qualitatively and quantitatively analyze the behavior and performance of our TabPFN on different tasks and compare it to previous approaches for tabular classification on 18 small, numerical datasets (see Section 5). Quantitatively, the TabPFN yields much better performance than any individual "base-level" classification algorithm, such as gradient-boosting via XGBoost (Chen and Guestrin, 2016), LightGBM (Ke et al., 2017) and CatBoost (Prokhorenkova et al., 2018), and in less than a second yields performance competitive to what the best available AutoML frameworks (Erickson et al., 2020; Feurer et al., 2021) achieve in one hour. Our in-depth qualitative analysis shows the TabPFN's predictions to be smooth and intuitive. Yet, its errors are quite uncorrelated to the errors of existing approaches, allowing additional performance improvements by ensembling. We also validate TabPFN's performance on an additional 67 datasets from OpenML Vanschoren et al. (2014).

We expect the revolutionary character of our claims to be met with initial skepticism and thus open-source all our code and the pre-trained TabPFN for scrutinization by the community, along with a scikit-learn-like interface, a Colab notebook and two online demos. Please see Section 8 for the links.

## 2 BACKGROUND ON PRIOR-DATA FITTED NETWORKS (PFNs)

First, we summarize how PFNs work; we refer to Müller et al. (2022) for the full details.

**The Posterior Predictive Distribution for Supervised Learning** In the Bayesian framework for supervised learning, the prior defines a space of hypotheses $\Phi$ on the relationship of a set of inputs $x$ to the output labels $y$. Each hypothesis $\phi \in \Phi$ can be seen as a mechanism that generates a data distribution from which we can draw samples forming a dataset. For example, given a prior based on structural causal models, $\Phi$ is the space of structural causal models, a hypothesis $\phi$ is one specific SCM, and a dataset comprises samples generated through this SCM. In practice, a dataset comprises training data with observed labels and test data where labels are missing or held out to assess predictive performance. The PPD for a test sample $x_{test}$ specifies the distribution of its label $p(\cdot|x_{test}, D_{train})$, which is conditioned on the set of training samples $D_{train} := \{(x_1, y_1), \ldots, (x_n, y_n)\}$. The PPD can be obtained by integration over the space of hypotheses $\Phi$, where the weight of a hypothesis $\phi \in \Phi$ is determined by its prior probability $p(\phi)$ and the likelihood $p(D|\phi)$ of the data $D$ given $\phi$:

$$p(y|x, D) \propto \int_{\Phi} p(y|x, \phi)p(D|\phi)p(\phi)d\phi. \tag{1}$$

**Synthetic Prior-fitting** Prior-fitting is the training of a PFN to approximate the PPD and thus do Bayesian prediction. We implement it with a prior which is specified by a prior sampling scheme of the form $p(D) = \mathbb{E}_{\phi \sim p(\phi)}[p(D|\phi)]$, which first samples hypotheses (generating mechanisms) with $\phi \sim p(\phi)$ and then synthetic datasets with $D \sim p(D|\phi)$. We repeatedly sample such synthetic datasets $D := (x_i, y_i)_{i \in \{1, \ldots, n\}}$ and optimize the PFN's parameters $\theta$ to make predictions for $D_{test} \subset D$, conditioned on the rest of the dataset $D_{train} = D \setminus D_{test}$. The loss of the PFN training thus is the cross-entropy on held-out examples of synthetic datasets. For a single test point $\{(x_{test}, y_{test})\} = D_{test}$, the training loss can be written as

$$\mathcal{L}_{PFN} = \mathbb{E}_{(\{(x_{test}, y_{test})\} \cup D_{train}) \sim p(D)}[-\log q_\theta(y_{test}|x_{test}, D_{train})]. \tag{2}$$

As shown by Müller et al. (2022), minimizing this loss approximates the true Bayesian posterior predictive distribution. We visualize this in Figure 1a and detail the full training setup in Algorithm 1 in the appendix. Crucially, this *synthetic prior-fitting* phase is performed only once for a given prior $p(D)$ as part of algorithm development.

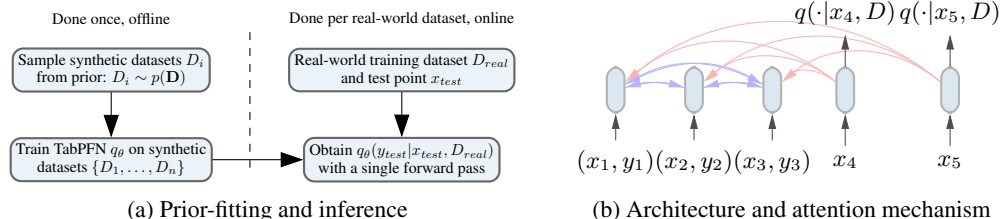

(a) Prior-fitting and inference  (b) Architecture and attention mechanism

Figure 1: Left (a): The PFN learns to approximate the PPD of a given prior in the offline stage to yield predictions on a new dataset in a single forward pass in the online stage. Right (b): Training samples $\{(x_1, y_1), \ldots, (x_3, y_3)\}$ are transformed to 3 tokens, which attend to each other; test samples $x_4$ and $x_5$ attend only to the training samples. Plots based on Müller et al. (2022).

**Real-World Inference** During inference, the trained model is applied to unseen real-world datasets. For a novel dataset with training samples $D_{train}$ and test features $x_{test}$, feeding $\langle D_{train}, x_{test} \rangle$ as an input to the model trained above yields the PPD $q_\theta(y|x_{test}, D_{train})$ in a single forward-pass. The PPD class probabilities are then used as predictions for our real-world task. Thus, PFNs perform training and prediction in one step (similar to prediction with Gaussian Processes) and do not use gradient-based learning on data seen at inference time.

**Architecture** PFNs rely on a Transformer (Vaswani et al., 2017) that encodes each feature vector and label as a token, allowing token representations to attend to each other, as depicted in Figure 1b. They accept a variable length training set $D_{train}$ of feature and label vectors (treated as a set-valued input to exploit permutation invariance) as well as a variable length query set of feature vectors $x_{test} = \{x_{(test,1)}, \ldots, x_{(test,m)}\}$ and return estimates of the PPD for each query.

**Tabular Data** Amongst other experiments, Müller et al. (2022) demonstrated PPD approximation with PFNs on binary classification for tiny, balanced tabular datasets with 30 training examples. Here, we substantially improve performance and scale up to small datasets with up to 1 000 data points and up to 10 classes, including imbalance. We discuss detailed changes in Appendix C.2.6.

## 3 THE TABPFN: A PFN FITTED ON A NEW PRIOR FOR TABULAR DATA

Our TabPFN is a Prior-data Fitted Network (PFN, see Section 2) that is fitted on data sampled from a novel prior for tabular data we introduce in Section 4. We modify the original PFN architecture (Müller et al., 2022) in two ways. i) We make slight modifications to the attention masks, yielding shorter inference times. ii) Additionally, we enable our model to work on datasets with different numbers of features by zero-padding. These architectural enhancements alongside our main contributions compared to Müller et al. (2022) are detailed in Appendix E.2.

In the prior-fitting phase, we train the TabPFN once on samples from the prior described in Section 4. To be more precise, we trained a 12-layer Transformer for 18 000 batches of 512 synthetically generated datasets each, which required a total of 20 hours on one machine with 8 GPUs (Nvidia RTX 2080 Ti). This yielded a single network that is used for all our evaluations. While this training step is moderately expensive, it is done offline, in advance, and only once for the TabPFN, as part of our algorithm development. The same TabPFN model is used for all experiments in this paper. Full details about our TabPFN training are given in Appendix E.

During inference, the TabPFN approximates the PPD for our dataset prior, i.e., it approximates the marginal predictions across our spaces of SCMs and BNNs (see Section 4), including a bias towards simple and causal explanations for the data. In our experiments, we present predictions for a single forward pass of our TabPFN, as well as predictions that ensemble 32 forward passes of datasets modified by a power transformation (applied with probability 0.5) and rotating the indices of feature columns and class labels (see Appendix C.2.6 for details).[1]

---

[1]Since we already rotate the indices of feature columns and class labels while training the TabPFN, it can in principle learn a representation that is invariant to both of these rotations. We expect that, using a longer training time or a larger Transformer, TabPFN would learn these invariances and make these rotations obsolete.

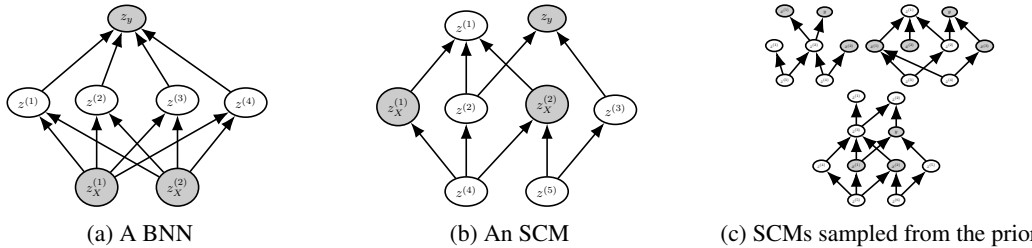

(a) A BNN          (b) An SCM          (c) SCMs sampled from the prior

Figure 2: Overview of graphs generating data in our prior. Inputs $x$ are mapped to the output $y$ through unobserved nodes $z$. Plots based on Müller et al. (2022).

## 4  A PRIOR FOR TABULAR DATA

The performance of our method crucially depends on the specification of a suitable prior, as the PFN approximates the PPD for this prior. Section 4.1 outlines a fundamental technique for our prior: we use distributions instead of point-estimates for almost all of our prior's hyperparameters. Section 4.2 motivates simplicity in our prior, while Sections 4.3 and 4.4 describe how we use Structural Causal Models (SCMs) and Bayesian Neural Networks (BNNs) as fundamental mechanisms to generate diverse data in our prior. Since our SCM and BNN priors only yield regression tasks, we show how to convert them to classification tasks in Section 4.5. We describe additional refinements to our prior to reflect peculiarities of tabular data (correlated and categorical features, exponentially scaled data and missing values) better in Appendix C.2.

### 4.1  FUNDAMENTALLY PROBABILISTIC MODELS

Fitting a model typically requires finding suitable hyperparameters, e.g., the embedding size, number of layers and activation function for NNs. Commonly, resource-intensive searches need to be employed to find suitable hyperparameters (Zoph and Le, 2017; Feurer and Hutter, 2019). The result of these searches, though, is only a point estimate of the hyperparameter choice. Ensembling over multiple architectures and hyperparameter settings can yield a rough approximation to a distribution over these hyperparameters and has been shown to improve performance (Zaidi et al., 2021; Wenzel et al., 2020). This, however, scales linearly in cost with the number of choices considered.

In contrast, PFNs allow us to be fully Bayesian about our prior's hyperparameters. By defining a probability distribution over the space of hyperparameters in the prior, such as BNN architectures, the PPD approximated by our TabPFN jointly integrates over this space and the respective model weights. We extend this approach to a mixture not only over hyperparameters but distinct priors: we mix a BNN and an SCM prior, each of which again entails a mixture of architectures and hyperparameters.

### 4.2  SIMPLICITY

We base our priors on a notion of simplicity, such as stated by Occam's Razor or the Speed Prior (Schmidhuber, 2002). When considering competing hypotheses, the simpler one is to be preferred. Work in cognitive science has also uncovered this preference for simple explanations in human thinking (Wojtowicz and DeDeo, 2020). Any notion of simplicity, however, depends on choosing a particular criterion that defines simplicity. In the following, we introduce priors based on SCMs and BNNs, in which we implement simplicity as graphs with few nodes and parameters.

### 4.3  SCM PRIOR

It has been demonstrated that causal knowledge can facilitate various ML tasks, including semi-supervised learning, transfer learning and out-of-distribution generalization (Schölkopf et al., 2012; Janzing, 2020; Rothenhäusler et al., 2018). Tabular data often exhibits causal relationships between columns, and causal mechanisms have been shown to be a strong prior in human reasoning (Waldmann and Hagmayer, 2013; Wojtowicz and DeDeo, 2020). Thus, we base our TabPFN prior on SCMs that model causal relationships (Pearl, 2009; Peters et al., 2017). An SCM consists of a collection $Z := (\{z_1, \ldots, z_k\})$ of structural assignments (called mechanisms): $z_i = f_i(z_{\mathrm{PA}_{\mathcal{G}}(i)}, \epsilon_i)$, where

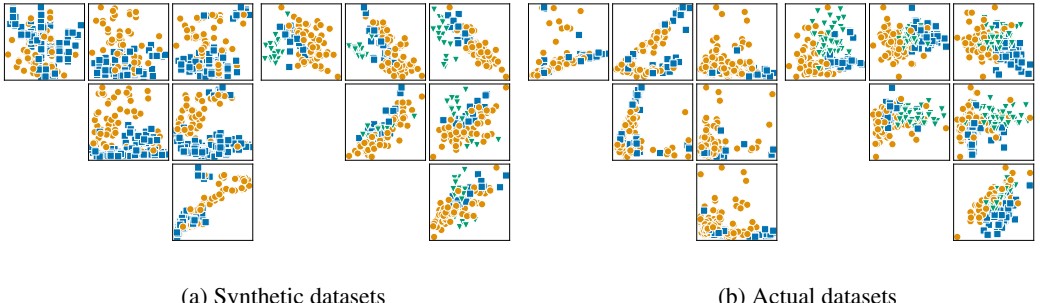

(a) Synthetic datasets                                    (b) Actual datasets

Figure 3: Each sub-plot shows the combination of two features, each dot represents a sample, color indicates the class label. (a) Two synthetic datasets generated by our causal tabular data prior. Numeric SCM outputs are mapped to classes as described in Section 4.5. (b) Two datasets from our validation datasets: Parkinsons (Left) and Wine (Right).

$\text{PA}_{\mathcal{G}}(i)$ is the set of parents of the node $i$ (its direct causes) in an underlying DAG $\mathcal{G}$ (the causal graph), $f_i$ is a (potentially non-linear) deterministic function and $\epsilon_i$ is a noise variable. Causal relationships in $\mathcal{G}$ are represented by directed edges pointing from causes to effects and each mechanism $z_i$ is assigned to a node in $\mathcal{G}$, as visualized in Figure 2.

**Predictions based on ideas from causal reasoning** Previous works have applied causal reasoning to predict observations on unseen data by using causal inference, a method which seeks to identify causal relations between the components of a system by the use of interventions and observational data (Pearl, 2010; Pearl and Mackenzie, 2018; Lin et al., 2021). The predicted causal representations are then used to make observational predictions on novel samples or to provide explainability. Most existing work focuses on determining a single causal graph to use for downstream prediction, which can be problematic since most kinds of SCMs are non-identifiable without interventional data, and the number of compatible DAGs explodes due to the combinatorial nature of the space of DAGs. In contrast, our TabPFN considers a broad family of SCMs but without any causal guarantees. We skip any explicit graph representation in our inference step and approximate the PPD directly. Thus, we do not perform causal inference but solve the downstream prediction task directly. This implicit assumption of SCM-like processes generating our data can be explained in Pearl's "ladder of causation", an abstraction of inference categories, where each higher rung represents a more involved notion of inference (Pearl and Mackenzie, 2018). At the lowest rung lies association, which includes most of ML. The second rung considers predicting the effect of interventions, i.e., what happens when we influence features directly. Our work can be considered as "rung 1.5", similar to Kyono et al. (2020; 2021): we do not perform causal reasoning, but make association-based predictions on observational data assuming SCMs model common datasets well. In Figure 8 in Appendix B, we experimentally show that TabPFN's predictions indeed align with simple SCM hypotheses.

**Defining a prior based on causal models** To create a PFN prior based on SCMs, we have to define a sampling procedure that creates supervised learning tasks (i.e., datasets). Here, each dataset is based on one randomly-sampled SCM (including the DAG structure and deterministic functions $f_i$). Given an SCM, we sample a set $z_X$ of nodes in the causal graph $\mathcal{G}$, one for each feature in our synthetic dataset, as well as one node $z_y$ from $\mathcal{G}$. These nodes are observed nodes: values of $z_X$ will be included in the set of features, while values from $z_y$ will act as targets. For each such SCM and list of nodes $z_X$ and $z_y$, $n$ samples are generated by sampling all noise variables in the SCM $n$ times, propagating these through the graph and retrieving the values at the nodes $z_X$ and $z_y$ for all $n$ samples. Figure 2b depicts an SCM with observed feature- and target-nodes in grey. The resulting features and targets are correlated through the generating DAG structure. This leads to features conditionally dependent through forward and backward causation, i.e., targets might be a cause or an effect of features. In Figure 3, we compare samples generated by two distinct SCMs to actual datasets, demonstrating the diversity in the space of datasets our prior can model.

In this work, we instantiate a large subfamily of DAGs and deterministic functions $f_i$ to build SCMs described in Appendix C.1. Since efficient sampling is the only requirement we have, the instantiated subfamily is very general, including multiple activation functions and noise distributions.

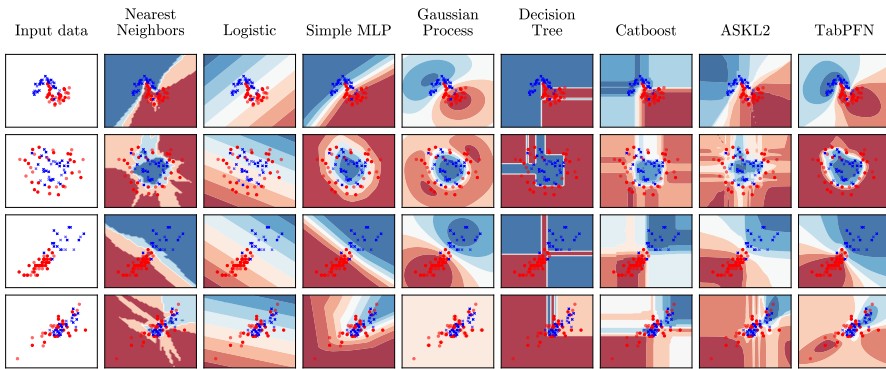

Figure 4: Decision boundaries on toy datasets generated with *scikit-learn* (Pedregosa et al., 2011).

## 4.4 BNN PRIOR

We also consider a BNN prior as introduced by Müller et al. (2022) and mix it with the SCM prior described above by randomly sampling datasets during PFN training from either one or the other prior with equal probability. To sample a dataset from the BNN prior, we first sample an NN architecture and its weights. Then, for each data point in the to-be-generated dataset, we sample an input $x$, feed it through the BNN with sampled noise variables and use the output $y$ as a target (see Figure 2a). An experimental ablation of this BNN prior to our final mixture of both SCM-based and BNN-based priors in Appendix B.4 demonstrates the strength of our novel SCM-based prior.

## 4.5 MULTI-CLASS PREDICTION

So far, the described priors return scalar labels. In order to generate synthetic classification labels for imbalanced multi-class datasets, we need to transform our scalar labels $\hat{y}$ to discrete class labels $y$. We do so by splitting the values of $\hat{y}$ into intervals that map to class labels:

i) We sample the number of classes $N_c \sim p(N_c)$, where $p(N_c)$ is a distribution over integers.
ii) We sample $N_c - 1$ class bounds $B_i$ randomly from the set of continuous targets $\hat{y}$.
iii) We map each scalar label $\hat{y}_i$ to the index of the unique interval that contains it: $y_i \leftarrow \sum_j [B_j < \hat{y}_i]$, where $[\cdot]$ is the indicator function.

For example, with $N_c = 3$ classes the bounds $B_c = \{-0.1, 0.5\}$ would define three intervals $\{(-\infty, -0.1], (-0.1, 0.5], (0.5, \infty)\}$. Any $\hat{y}_i$ would be mapped to the label 0 if it is smaller than $-0.1$, to 1 if lies in $(-0.1, 0.5]$ and to 2 otherwise. Finally, we shuffle the labels of classes, i.e. we remove the ordering of class labels w.r.t. the ranges.

## 5 EXPERIMENTS

### 5.1 EVALUATION ON TOY PROBLEMS

We first qualitatively compare our TabPFN to standard classifiers (without hyperparameter tuning) in Figure 4. The top row shows the *moons* dataset with noise. The TabPFN accurately models the decision boundary between samples; also, similar to Gaussian processes, uncertainties are large for points far from observed samples. The second row shows the *circles* dataset with noise: the TabPFN accurately models the circle's shape with high confidence anywhere outside the region where samples are mixed. The third row shows two classes and features from the iris dataset, while the fourth rows shows two classes and features from the wine dataset (both in *scikit-learn* (Pedregosa et al., 2011)); in both of these cases, the TabPFN makes intuitive, well-calibrated predictions.

### 5.2 EVALUATION ON TABULAR ML TASKS

Now, we turn to an empirical analysis of our method for real-world classification tasks. We compare our method against state-of-the-art ML and AutoML methods for tabular classification.

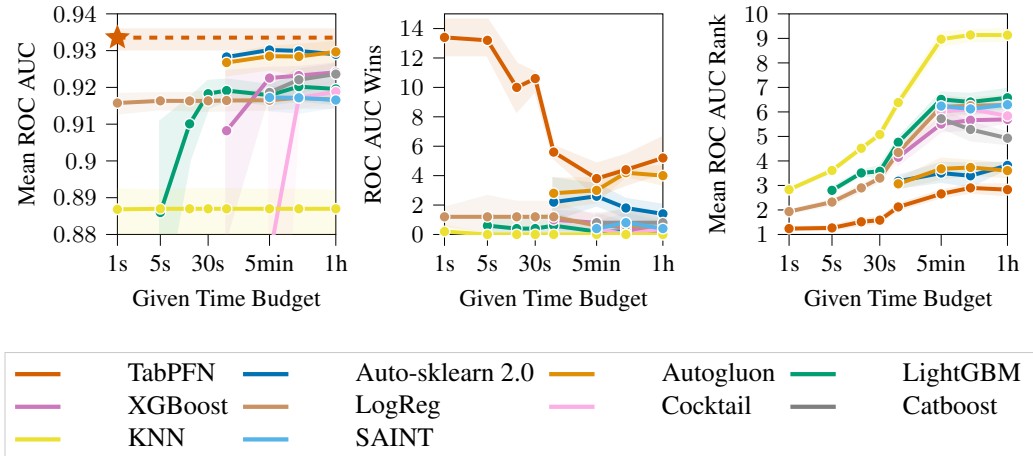

Figure 5: ROC AUC as a function of the time allowed to train & tune methods, on 18 numerical datasets from the OpenML-CC18 Benchmark. We report the mean, mean wins and rank and 95% confidence interval across 5 splits for increasing budgets. The red star indicates performance of TabPFN with 32 permutations (which requires 0.62s on GPU). We report more results in Table 1.

**Datasets** As test datasets, we used all datasets from the curated open-source OpenML-CC18 benchmark suite (Bischl et al., 2021) that contain up to 2 000 samples (1 000 for the training split), 100 features and 10 classes. The resulting set comprises 30 datasets. We split these datasets into 18 datasets that contain only numerical features and no missing values, and 12 other datasets that contain categorical features and/or missing values. In the main paper, in Figure 5 and Table 1, we limit our analysis to the case of numerical datasets without missing values, which we focussed the development of our TabPFN prior on. Results on all of the 30 datasets are given in Appendix B.1 and still show strong aggregate performance for TabPFN, albeit not as strong as for the purely numerical case, due to generally worse performance of TabPFN on datasets with categorical features and/or missing values. We focus on small datasets because (1) small datasets are often encountered in real-world applications (Dua and Graff, 2017), (2) existing DL methods are most limited in this domain (Grinsztajn et al., 2022) and (3) the TabPFN would be significantly more expensive to train and evaluate for larger datasets, a limitation detailed in Appendix A.

**Baselines** We compare against five standard ML methods and two state-of-the-art AutoML systems for tabular data. As ML models we considered two simple and fast baselines, *K-nearest-neighbors (KNN)* and *Logistic Regression (LogReg)*. Additionally, we considered three popular tree-based boosting methods, *XGBoost* (Chen et al., 2020), *LightGBM* (Ke et al., 2017) and *CatBoost* (Prokhorenkova et al., 2018). For each ML model, we used 5-fold cross-validation to evaluate randomly drawn hyperparameter configurations until a given budget was exhausted or 10 000 configurations were evaluated (for the search spaces, see Appendix F.2). We then chose the best-performing hyperparameter configuration (maximum ROC AUC OVO) and refit on the whole training set. Where necessary, we imputed missing values with the mean, one-hot or ordinal encoded categorical inputs, normalized features and passed categorical feature indicators. As more complex but powerful baselines, we chose two state-of-the-art AutoML systems: *AutoGluon* (Erickson et al., 2020), which combines ML models including neural networks and tree-based models into a stacked ensemble, and *Auto-sklearn 2.0* (Feurer et al., 2015; 2021) which uses Bayesian Optimization and combines the evaluated models into a weighted ensemble.[2] We note that previous works have found that DL baselines do not outperform or match the performance of GBDT or AutoML methods for small to medium-sized tabular data (< 10,000 samples; while matching GBDT performance on larger datasets) (Borisov et al., 2021; Grinsztajn et al., 2022; Shwartz-Ziv and Armon, 2022). Furthermore, DL methods such as TabNet, SAINT, Regularization Cocktails, Non-parametric Transformers (Arik and Pfister, 2021; Somepalli et al., 2021; Kadra et al., 2021; Kossen et al., 2021) are evaluated on much larger

---

[2]Auto-sklearn 2.0 optimizes ROC AUC for binary classification and cross-entropy for multi-class classification (as multi-class ROC AUC is not implemented).

|  | LightGBM | CatBoost | XGBoost | ASKL2.0 | AutoGluon | TabPFN$_{n.e.}$ | TabPFN | TabPFN + AutoGluon |
|---|---|---|---|---|---|---|---|---|
| M. rank AUC OVO | 6.9722 | 4.9444 | 6.1944 | 4.4722 | 4 | 3.8056 | 2.9444 | **2.6667** |
| Mean rank Acc. | 6.8889 | 4.9722 | 6.0556 | 5.1667 | 3.8889 | 3.8889 | 2.8889 | **2.25** |
| Mean rank CE | 5.7778 | 5.4444 | 6 | 6.4167 | 3.1111 | 4.1389 | 3.0278 | **2.0833** |
| Mean AUC OVO | 0.92±.013 | 0.924±.011 | 0.924±.01 | 0.929±.0096 | 0.93±.0091 | 0.932±.0088 | **0.934**±.0086 | **0.934**±.0084 |
| Mean Acc. | 0.862±.012 | 0.864±.011 | 0.866±.011 | 0.87±.014 | 0.881±.01 | 0.873±.0095 | 0.879±.0089 | **0.886**±.0094 |
| Mean CE | 0.75±.039 | 0.747±.029 | 0.759±.04 | 0.813±.073 | 0.714±.014 | 0.727±.021 | 0.716±.019 | **0.711**±.014 |
| Mean time (s) (Tune + Train + Predict) | 3280 | 3746 | 3364 | 3601 | 3077 | **1.301** (CPU) **0.0519** (GPU) | 37.59 (CPU) 0.6172 (GPU) | 3109 (CPU) 3077 (GPU) |

Table 1: Results on the 18 numerical datasets in OpenML-CC18 for 60 minutes requested time per data-split. ± values indicate a metric's standard deviation. If available, all baselines optimize ROC AUC. TabPFN n.e. is a faster version of our method, without ensembling, while TabPFN + AutoGluon is an ensemble of TabPFN and AutoGluon. Separate inference times in Table 2.

datasets and often use custom parameter tuning and preprocessing. We still evaluate two prominent DL methods: *Regularization Cocktails*[3] (Kadra et al., 2021) and *SAINT* (Somepalli et al., 2021).

**Evaluation Protocol**   For each dataset and method, we evaluated 5 repetitions, each with a different random seed and train- and test split (50% train and 50% test samples; all methods used the same split given a seed). To aggregate results across datasets, we report the ROC AUC (one-vs-one (OVO) for multi-class classification) average, ranks and wins including the 95% confidence interval and compare to the performance of the baselines with a budget of $\{30, 60, 300, 900, 3600\}$ seconds.[4] Our TabPFN uses 32 data permutations for ensembling as described in Section 3; we also evaluate TabPFN without permutations, which we label "TabPFN$_{n.e.}$" in Table 1.

**Results**   We now present results for the 18 purely numerical datasets without missing values, detailed in Figure 5 and Table 1. Figure 5 shows that TabPFN achieves a dramatically better tradeoff of accuracy and training speed than all the other methods: it makes predictions within less than a second on one GPU that tie with the performance of the best competitors (the AutoML systems) after training one hour, and that dominate the performance of tuned GBDT methods. Unsurprisingly, the simple baselines (*LogReg*, *KNN*) already yield results with a small budget but perform worst overall. GBDTs (*XGBoost*, *CatBoost*, *LightGBM*) perform better but are still outperformed by TabPFN and the state-of-the-art AutoML systems (*Auto-sklearn 2.0*, *Autogluon*).

TabPFN is much faster than methods with comparable performance. In the following, we compare the combined times for training and prediction (and tuning if applicable). TabPFN$_{n.e.}$ requires 1.30s on a CPU and 0.05s on a GPU on average to predict for one dataset[5], performing comparably to the strongest baselines at five minutes; it thus yields a 230× speedup on CPU and a 5 700× speedup using a GPU. We ignore computational development costs of each method, see Appendix F.5 for why.

We would like to emphasize that the discussed results are *aggregate* results across datasets, and that no classification method, including TabPFN, performs best on all individual datasets. Indeed, there do exist datasets for which TabPFN is outperformed even by default baselines. Generally, TabPFN is less strong when categorical features or missing values are present. Appendix B.1 presents results for all 30 test datasets from the OpenML-CC18 Benchmark, including 12 with categorical features and/or missing values. Appendix B.5 presents a more detailed overview of the results for different kinds of datasets, including on an additional 149 validation datasets from OpenML, confirming TabPFN's strong performance for purely numerical datasets. We also show per dataset results for each of the 179 datasets (our 30 test datasets and the 149 validation datasets); these demonstrate that, while TabPFN generally does better for datasets with numerical features, there also exist several purely numerical datasets for which TabPFN does *not* perform better than the baselines, as well as categorical datasets for which it *does* clearly perform best.

---

[3]The implementation of Regularization Cocktails does not currently support optimization for multi-class AUC ROC. We optimized for cross-entropy, which performed better than balanced accuracy (the default).

[4]When comparing methods to each other for a given time budget in Figure 5, we drop methods that take more than 200% of the requested time budget; some methods also do not use their full budget.

[5]This includes overhead from a conservative scikit-learn interface. We assume that our trained model is in (GPU) memory already, which otherwise required an additional 0.2s for us.

**OpenML-AutoML Benchmark**    Additionally, we evaluated our TabPFN on the small ($\leq 1\,000$ training samples, 100 features, 10 classes) datasets of the OpenML-AutoML Benchmark, for which externally-validated performance numbers are available. We use the setup provided by the OpenML-AutoML Benchmark, i.e.: metrics, official evaluation scripts and pre-released evaluations for an even wider range of AutoML baselines. Using only an average of $4.4$ seconds per dataset on a single CPU, compared to 60 minutes for the baselines, TabPFN outperformed all baselines in terms of mean cross-entropy, accuracy and the OpenML Metric [6]. We provide detailed results in Appendix B.2.

**In-depth analysis of TabPFN predictions**    We evaluated our model predictions in a plethora of ways to provide extra insights, such as the inductive biases of our method, see Appendix B.3. We confirm that TabPFN learns to make predictions biased towards simple causal explanations, as detailed in B.3.1, while GBDT methods do not share this inductive bias. We also evaluate our method's invariance to feature rotations and robustness to uninformative features in Appendix B.3.2 and B.3.3. In Figure 6, we observe that TabPFN is especially strong compared to baselines when datasets do not contain categorical features or missing values.

**Ensembling**    We observe that TabPFN performs best on different datasets than our baselines, i.e., the per-dataset correlation of normalized ROC AUC scores between TabPFN and the strong baselines is lower than between the strong baselines, visualized in Figure 11 in our Appendix. This is likely due to the novel inductive biases of our approach, which lead to distinct predictions. Ensembling predictions is more effective when the considered methods make fewer correlated errors (Breimann, 2001), which encourages the use of more diverse strategies, making TabPFN an ideal candidate for ensembling with baseline methods (Wu et al., 2021). Also, TabPFN is evaluated so quickly, that predictions are almost free, compared to the long runtimes of the baselines. To demonstrate its potential for ensembling, we include an entry *"TabPFN + AutoGluon"* in Table 1, generated by averaging the predictions of TabPFN and *AutoGluon*; this strongly outperforms all other methods.

**Model Generalization**    The PFN architecture accepts datasets of any length as input. However, when training our model in the *synthetic prior-fitting* phase, we limited synthetic data to a maximum size of $1\,024$, as prior-fitting becomes more expensive with larger datasets. Nevertheless, we wondered: Would TabPFN generalize to larger training set sizes that were never seen during training? To test this, we used a collection of 18 datasets from the OpenML-AutoML Benchmark (see Table 11) from which we selected $10\,000$ samples each. We evaluated TabPFN on up to $5\,000$ training samples, using 5 random data splits and $5000$ test samples. Surprisingly, our models generalize beyond sample sizes seen during training, as shown in Figure 10 in Appendix B.1.

## 6    CONCLUSIONS & FUTURE WORK

We have shown how a single Transformer, the TabPFN, can be trained to do the work of a full AutoML framework for tabular data and can yield predictions in 0.4 seconds that are competitive with the performance that the best available AutoML frameworks achieve in $5$ to $60$ minutes. This slashes the computational expense of AutoML, enabling affordable, green, AutoML.

The TabPFN still has important limitations: the underlying Transformer architecture only scales to small datasets as detailed in Appendix A; our evaluations focused on classification datasets with only up to $1\,000$ training samples, 100 purely numerical features without missing values and 10 classes, which motivates work on (1) scaling up to large datasets. Our in-depth analysis of the inductive biases of TabPFN (see Appendix B.3), points towards extensions for (2) improved handling of categorical features, (3) missing values and (4) robustness to unimportant features. Also, our work motivates a multitude of exciting follow-ups regarding (5) integration of TabPFN into existing AutoML frameworks; (6) ensembling to continue making improvements given more time; (7) dataset-dependent choices of the prior; (8) generalizations to non-tabular data and (9) regression tasks, as well as studies of the TabPFN regarding the dimensions of trustworthy AI (e.g., (10) out-of-distribution robustness; (11) algorithmic fairness, (12) robustness to adversarial examples, and (13) explainability). The almost instant state-of-the-art predictions of TabPFN are also likely to give rise to (14) novel exploratory data analysis methods, (15) novel feature engineering methods and (16) novel active learning methods. Finally, our advances in causal reasoning warrant follow-ups on (17) approximating the effects of interventions and counterfactuals considering a distribution of SCMs.

---

[6]The OpenML metric evaluates binary classification using ROC AUC and multiclass using Cross Entropy.

## 7 ETHICS STATEMENTS

In terms of broader societal impact of this work, we do not see any foreseeable strongly negative impacts. However, this paper could positively impact the carbon footprint and accessibility of learning algorithms. The computations required for machine learning research have been doubling every few months, resulting in a large carbon footprint (Schwartz et al., 2020). Moreover, the financial cost of the computations can make it difficult for academics, students, and researchers to apply these methods. The decreased computational time shown by TabPFN translates to reductions in $CO2$ emissions and cost, making it available to an audience that does not have access to larger scale computing.

As the TabPFN provides a highly portable and convenient way of building new classifiers that work in real-time, it is likely to increase the pervasiveness of machine learning even further. While this can have many positive effects on society, such as better personalized healthcare, increased customer satisfaction, efficiency of processes, etc, it will also be crucial to study and improve the TabPFN under the lens of the many dimensions of trustworthy AI other than computational sustainability, such as algorithmic fairness, robustness to adversarial examples, explainability, and auditability. We hope that its foundation in causal models and simplicity will allow possible avenues for work along these lines.

## 8 REPRODUCIBILITY

**Code release** In an effort to ensure reproducibility, we release code alongside our pre-trained TabPFN and notebooks to reproduce our experiments at `https://github.com/automl/TabPFN`.

**Application to public benchmarks** In our work, we evaluate TabPFN to publicly available benchmarks: the OpenML-CC18 Benchmark and the OpenML-AutoML Benchmark. This ensures, that dataset choices are not cherry-picked to our method. Also, for the OpenML-AutoML Benchmark, we use official baseline results and evaluate our method using the evaluation scripts published for this benchmark.[7]

**Availability of datasets** All datasets used in our experiments are freely available at `OpenML.org` (Vanschoren et al., 2014), with downloading procedures included in the submission. Further details on the datasets used can be found in Section F.3.

**Online resources** We created a Colab notebook, that lets you interact with our scikit-learn interface at `https://colab.research.google.com/drive/1J0l1AtMV_H1KQ7IRbgJje5hMhKHczH7-?usp=sharing`.

We created another Colab, where our evaluation and plots on 179 test and validation datasets can be reproduced easily. `https://colab.research.google.com/drive/1yUGaAf3D7RSyO5Jc4PYXSVbtaUAozh6S`

We also created two demos. One to experiment with the TabPFNs predictions (`https://huggingface.co/spaces/TabPFN/TabPFNPrediction`) and one to check cross-validation ROC AUC scores on new datasets (`https://huggingface.co/spaces/TabPFN/TabPFNEvaluation`). Both of them run on a weak CPU, thus it can require a little bit of time.

**Details of training procedures for TabPFN and baselines** Details shared in the training procedure of all our Transformer models can be found in Appendix F. An overview of the hyperparameters used for running the TabPFN and our baselines can be found in Tables 5 and 6, respectively.

ACKNOWLEDGMENTS

Robert Bosch GmbH is acknowledged for financial support. This research was supported by the Deutsche Forschungsgemeinschaft (DFG, German Research Foundation) under grant number 417962828, the state of Baden-Württemberg through bwHPC and the German Research Foundation (DFG) through grant no INST 39/963-1 FUGG, and TAILOR, a project funded by EU Horizon 2020 research, and innovation programme under GA No 952215. We acknowledge funding through the

---

[7]Available at https://github.com/openml/automlbenchmark

European Research Council (ERC) Consolidator Grant "Deep Learning 2.0" (grant no. 101045765). Funded by the European Union. Views and opinions expressed are however those of the author(s) only and do not necessarily reflect those of the European Union or the ERC. Neither the European Union nor the ERC can be held responsible for them.

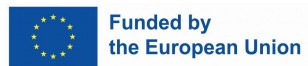

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

## A    LIMITATIONS

The runtime and memory usage of the Transformer-based PFN architecture used in this work scales quadratically with the number of inputs, i.e., training samples passed. Thus, inference on larger sequences (> 100 000) is hard on current consumer GPUs. A growing number of methods seek to tackle this issue and report similar performances while scaling linearly with the number of inputs (Zaheer et al., 2020; Beltagy et al., 2020). These methods can be integrated into the PFN architecture and thus into the TabPFN. Furthermore, in our experiments we limit the number of features to 100 and the number of classes to 10 as described in Section 5. While this choice is flexible, the precise TabPFN that we fitted cannot work with datasets that go beyond these limits. We also focused the development of TabPFN to purely numerical datasets without missing values, and while they *can* be applied to datasets with categorical features and/or missing values, their performance is generally worse. We hope to tackle this problem in future versions of TabPFN by a modified architecture and prior. Finally, we did not consider the existence of many uninformative features in our prior, leading to performance degradation when such features are added; we hope to address this issue in future versions of TabPFN. While baseline models (except Gaussian Processes) fit and predict in separate steps, TabPFN performs both at the same time. Thus, baseline models are often faster than TabPFN in terms of pure inference time (see Table 2).

## B    ADDITIONAL RESULTS

### B.1    DETAILED TABULAR RESULTS

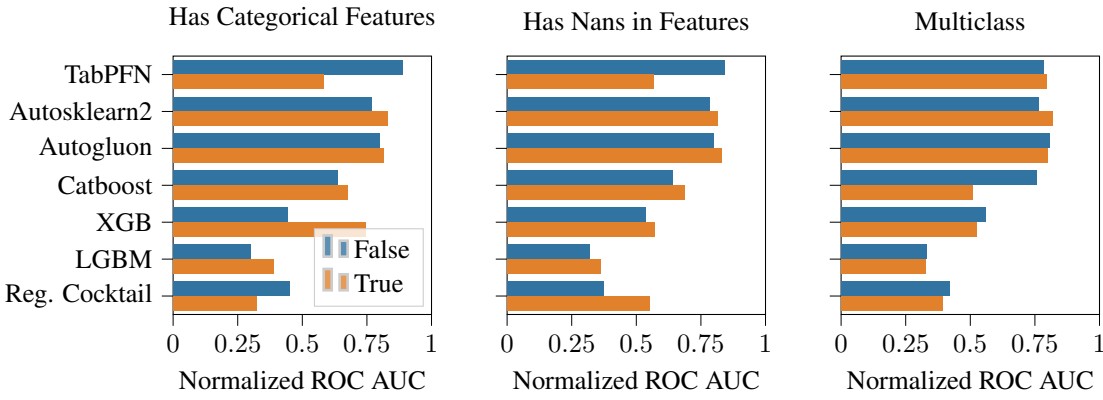

Figure 6: Normalized ROC AUC performance on datasets from the OpenML-CC18 Benchmark, divided by dataset characteristics. For each plot, we split the datasets into two groups. Left: Orange bars indicate the performance on datasets that have categorical features. Middle: Orange bars indicate datasets that contain missing values. Right: Orange bars indicate multiclass datasets, while others are binary.

In Figure 6, we explore how the kind of dataset evaluated affects the performance of TabPFN, compared to our baselines. We find that TabPFN performs much better when no categorical features are present. We also find that TabPFN performs better, when no missing values are present in the data. This warrants an extension of our prior in future work, to make it more customized towards categorical and missing data. Our method seems to work comparably well for binary and multi-class problems.

In addition to the results in the main paper in Section 5.2, we report detailed results on all 30 datasets in OpenML-CC18, which are small enough, but might include categorical features and missing values. We show performance over time in Figure 7 and a wide range of performance values and per dataset results with a 1 hour time limit in Table 2.

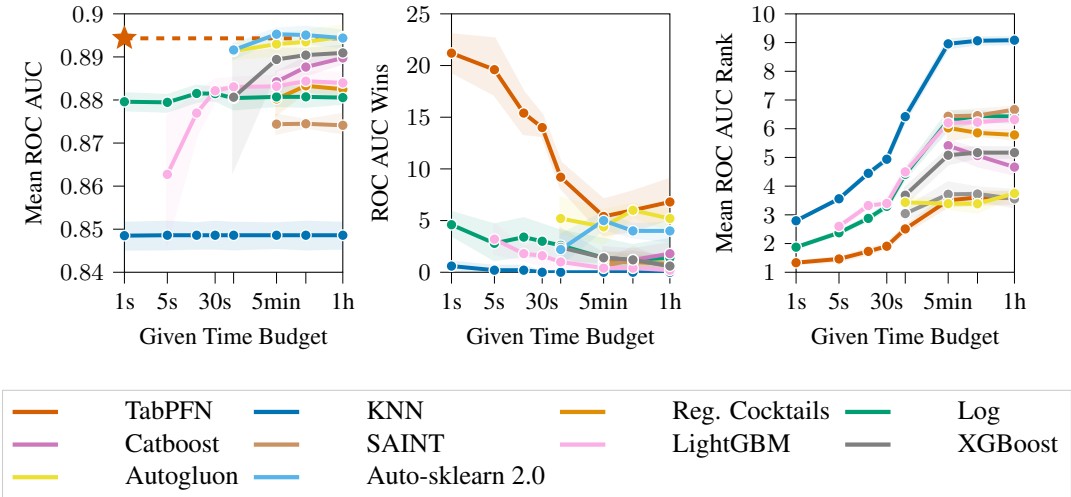

Figure 7: ROC AUC performance over time on the 30 small OpenML-CC18 including datasets with categorical features and missing values. We report the mean, mean wins and rank along with the 95% confidence interval across 5 splits for increasing training and tuning time budgets (Unlabelled ticks: 1min, 15min). The red star indicates performance of TabPFN with 32 data permutations (which requires 0.62s on GPU). We report detailed results with a 60 min budget in Table 1

## B.2 RESULTS ON THE OPENML-AUTOML BENCHMARK

We evaluate TabPFN using the official benchmarking scripts,[8] datasets, splits and baselines results of the OpenML-AutoML Benchmark. The full list of 5 datasets can be found in Table 10. We note that these datasets are not disjoint from our evaluation datasets; in fact 3 of these ("credit-g", "vehicle", and "blood-transfusion-service-center") were included in our evaluation datasets from the OpenML-CC18 Benchmark as well, while one dataset ("Australian") was included in our list of 150 meta-validation datasets. The evaluation using another set of benchmarking scripts with previously published train-test splits and baseline results, however, helps to confirm that: (1) Our baselines are well tuned and not outperformed by another method in the extensive OpenML-AutoML Benchmark; (2) the runtimes of TabPFN are reproducible in a controlled environment provided by the OpenML-AutoML Benchmark; and (3) TabPFN is not overfit to datasplits or our evaluation metric. While we used a 50-50 train-test split with 5 iterations for our experiments in Table 3, the OpenML-AutoML Benchmark uses a 10-fold cross-validation, which results in a 90-10 splits with 10 iterations. Thus in the OpenML-AutoML Benchmark, all methods use more training samples than in our experiments on the OpenML-CC18 Benchmark, which leads to slightly stronger results.

## B.3 IN-DEPTH ANALYSIS OF MODEL BIASES

Previous work by Grinsztajn et al. (2022) empirically investigates the inductive biases of tree-based and deep-learning models. They identify three challenges in developing tabular-specific models. Models must be (1) able to fit irregular functions; (2) robust to uninformative features; and (3) preserve the orientation of the data. We perform and extend this analysis in the following section. We consider three model types in our analyses: TabPFN, GBDTs (LightGBM) and NNs (Standard Sklearn Multi Layer Perceptron with a hidden dimensionality of 100). We do not seek to make absolute comparisons between these model types, which would warrant tuning of our baselines, but only seek to investigate their qualitative behavior in the following experiments.

### B.3.1 FITTING IRREGULAR PATTERNS

We believe GBDT methods tend to learn non-smooth and irregular patterns in the targets, while MLPs learn smooth, low-frequency functions, as can be seen in Figure 4. TabPFN learns target functions that are rather smooth, as suggested in Figure 4. This is due to our model's prior, which prefers

---

[8] Available at https://github.com/openml/automlbenchmark

Table 2: ROC AUC OVO results on the 30 small OpenML-CC18 (including datasets with categorical features and missing values) for 60 minutes requested time per dataset and per split. If available, all baselines are given ROC AUC optimization as an objective, others optimize CE. Overall each method got a time budget of 150 hours, but not all methods used the full budget. Times for TabPFN refer to times on GPU. TabPFN runs training and prediction in a joint step and does not have any hyperparameter tuning, so only a single aggregate time is shown.

| | LightGBM | CatBoost | XGBoost | ASKL2.0 | AutoGluon | TabPFN$_{n.e.}$ | TabPFN | TabPFN + AutoGluon |
|---|---|---|---|---|---|---|---|---|
| balance-scale | 0.9938 | 0.9245 | 0.9939 | 0.997 | 0.9919 | 0.9965 | **0.9973** | 0.9958 |
| mfeat-fourier | 0.9786 | 0.9816 | 0.9803 | 0.9826 | **0.9843** | 0.9767 | 0.9811 | 0.9838 |
| breast-w | 0.991 | 0.9931 | 0.9896 | 0.9939 | 0.9933 | 0.9931 | 0.9934 | **0.994** |
| mfeat-karhunen | 0.9979 | 0.9986 | 0.9983 | 0.9975 | **0.9987** | 0.9939 | 0.9978 | 0.9985 |
| mfeat-morphologica.. | 0.9601 | 0.9629 | 0.9612 | 0.9671 | 0.9698 | 0.9657 | 0.9669 | **0.9722** |
| mfeat-zernike | 0.9716 | 0.9759 | 0.9735 | 0.9812 | **0.9908** | 0.9812 | 0.9823 | 0.9901 |
| cmc | 0.7288 | 0.7256 | 0.7299 | **0.7378** | 0.7331 | 0.7233 | 0.7276 | 0.7336 |
| credit-approval | 0.9415 | 0.9389 | **0.9422** | 0.9406 | 0.9415 | 0.9253 | 0.9322 | 0.9394 |
| credit-g | 0.7684 | 0.7852 | 0.7853 | 0.793 | 0.7941 | 0.7894 | 0.7894 | **0.7948** |
| diabetes | 0.8247 | 0.8383 | 0.8378 | 0.8343 | 0.8391 | 0.8412 | 0.841 | **0.8427** |
| tic-tac-toe | 0.9988 | 0.9992 | 1 | 0.9943 | 1 | 0.9547 | 0.9759 | 0.9992 |
| vehicle | 0.9232 | 0.9302 | 0.9282 | 0.9504 | 0.9416 | 0.9568 | **0.9589** | 0.9538 |
| eucalyptus | 0.8931 | 0.8979 | 0.9004 | 0.9132 | 0.9204 | 0.9218 | 0.9245 | **0.9278** |
| analcatdata_author.. | 0.9999 | 0.9999 | 0.9997 | 1 | 0.9993 | 1 | 1 | 1 |
| analcatdata_dmft | 0.5461 | 0.5589 | 0.5743 | 0.5752 | 0.5657 | 0.5643 | **0.579** | 0.5756 |
| pc4 | 0.9301 | 0.9413 | 0.9291 | 0.9331 | 0.9428 | 0.9298 | 0.9383 | **0.944** |
| pc3 | 0.8178 | 0.8247 | 0.8288 | 0.8265 | 0.8282 | 0.8308 | **0.8373** | 0.836 |
| kc2 | 0.8141 | 0.8323 | 0.8227 | 0.8311 | 0.8242 | 0.8322 | **0.8346** | 0.8321 |
| pc1 | 0.8321 | 0.86 | 0.8489 | 0.8527 | 0.8578 | **0.877** | 0.8761 | 0.8739 |
| banknote-authentic.. | **1** | **1** | **1** | **1** | **1** | **1** | **1** | **1** |
| blood-transfusion-.. | 0.7144 | 0.7403 | 0.7312 | 0.7504 | 0.7364 | 0.753 | **0.7549** | 0.7469 |
| ilpd | 0.6917 | 0.7279 | 0.7171 | 0.7212 | 0.723 | **0.7412** | 0.7379 | 0.7326 |
| qsar-biodeg | 0.9126 | 0.9217 | 0.9191 | 0.9247 | 0.9276 | **0.9345** | 0.9336 | 0.9336 |
| wdbc | 0.9904 | 0.9931 | 0.9904 | 0.9947 | 0.9956 | 0.996 | **0.9964** | 0.996 |
| cylinder-bands | 0.8556 | 0.8757 | 0.8782 | 0.8718 | **0.8878** | 0.8314 | 0.8336 | 0.8751 |
| dresses-sales | 0.5593 | 0.5696 | **0.5823** | 0.5705 | 0.5507 | 0.5333 | 0.5376 | 0.5509 |
| MiceProtein | 0.9997 | 0.9999 | 0.9998 | 0.9999 | 1 | 0.9997 | 0.9999 | 1 |
| car | 0.9925 | 0.9955 | 0.9948 | **0.998** | 0.997 | 0.9926 | 0.995 | 0.9972 |
| steel-plates-fault.. | 0.9626 | 0.9655 | 0.9656 | **0.9694** | 0.9666 | 0.9619 | 0.9655 | 0.9687 |
| climate-model-simu.. | 0.9286 | 0.9344 | 0.9255 | 0.9291 | 0.9391 | **0.9426** | 0.9415 | 0.9421 |
| Wins AUC OVO | 0 | 0 | 2 | 2 | 2 | 4 | **5** | **5** |
| Wins Acc. | 0 | 2 | 2 | 3 | 3 | 0 | 6 | **8** |
| Wins CE | 0 | 1 | 3 | 1 | 7 | 1 | 6 | **9** |
| M. rank AUC OVO | 6.6167 | 4.9667 | 5.4167 | 4.05 | 3.7833 | 4.65 | 3.7 | **2.8167** |
| Mean rank Acc. | 6.5333 | 4.9833 | 5.1833 | 4.8667 | 3.8167 | 4.5333 | 3.6167 | **2.4667** |
| Mean rank CE | 5.7333 | 5.6 | 5.4667 | 5.8 | 2.8667 | 4.6167 | 3.5333 | **2.3833** |
| Win/T/L AUC vs Tab.. | 5/4/21 | 9/4/17 | 6/5/19 | 10/6/14 | 13/4/13 | 4/8/18 | –/–/– | 15/7/8 |
| Win/T/L Acc vs Tab.. | 6/0/24 | 9/1/20 | 11/0/19 | 11/2/17 | 12/0/18 | 6/3/21 | –/–/– | 19/3/8 |
| Win/T/L CE vs TabP.. | 6/0/24 | 8/0/22 | 8/0/22 | 8/0/22 | 20/0/10 | 1/4/25 | –/–/– | 23/0/7 |
| Mean AUC OVO | 0.884±.012 | 0.89±.011 | 0.891±.011 | 0.894±.01 | 0.895±.01 | 0.891±.01 | 0.894±.01 | **0.898±.0097** |
| Mean Acc. | 0.815±.014 | 0.818±.011 | 0.821±.013 | 0.821±.016 | 0.83±.012 | 0.82±.013 | 0.825±.012 | **0.834±.011** |
| Mean CE | 0.782±.074 | 0.767±.061 | 0.758±.047 | 0.815±.06 | **0.72±.015** | 0.742±.021 | 0.732±.018 | 0.721±.015 |
| Time Tune + Train (s) | 3241 | 3718 | 3304 | 3601 | 3127 | **0.0519** | 0.6172 | 3127 |
| Predict (s) | 0.0815 | 0.0168 | 0.0685 | 1.224 | 21.18 | | | |

simple SCMs as explanations and thus less irregular decision planes. We note, however, that when many training samples are provided, TabPFN also fits more complex functions.

The tradeoff between complexity and number of training samples is explored in Figure 8. Here, we evaluate the training set cross-entropy loss on synthetic data generated from random SCMs. The number of training samples and the complexity of the generated data (number of hidden units in the data generating graph) is varied.

### B.3.2 ROBUSTNESS TO UNINFORMATIVE FEATURES

Tabular datasets contain a large fraction of uninformative features (Grinsztajn et al., 2022). To evaluate robustness to uninformative features, we add an increasingly large fraction of uninformative features to our data and show results in Figure 9. Uninformative features are generated by copying existing features and shuffling their values randomly between samples. We find that TabPFN and MLPs are less robust to uninformative features than LightGBM. TabPFN could be adapted by including more uninformative features in the used prior. In a second experiment we drop an increasingly large fraction of features. We drop these features according to feature importance (ranked by a Random Forest), first removing least informative features. We show results in Figure 9 (middle) and observe that the classification accuracy of a TabPFN and GBDT is not much affected by removing up to

Table 3: ROC AUC OVO results on the 5 small datasets ($\leq 1\,111$ examples, 100 features and 10 classes) for 60 minutes requested time per dataset and per split. If available, all baselines are given ROC AUC optimization as an objective, others optimize CE. Evaluation on this benchmark was performed with our previously released TabPFN in order to mitigate test-set overfitting. Mean OpenML-Metric is not shown in the table as averaging over a mixture of Cross Entropy and ROC AUC is problematic (however, TabPFN had the strongest average as well).

| | AutoGluon | ASKL | ASKL2.0 | TunedRandomForest | FLAML | TPOT | TabPFN |
|---|---|---|---|---|---|---|---|
| vehicle | -0.3084 | -0.3816 | -0.3412 | -0.4849 | -0.4286 | -0.3433 | **-0.2955** |
| eucalyptus | -0.6905 | -0.7255 | -0.6967 | -0.7209 | -0.7433 | -0.7123 | **-0.665** |
| blood-transfusion-service-center | 0.7532 | 0.749 | 0.7557 | 0.6879 | 0.7332 | 0.7359 | **0.7593** |
| Australian | 0.941 | 0.9315 | **0.9411** | 0.9394 | 0.9356 | 0.9382 | 0.9395 |
| credit-g | 0.7977 | 0.7891 | 0.7984 | **0.8017** | 0.7838 | 0.7821 | 0.7989 |
| Wins OpenML Metric | 0 | 0 | 0 | 1 | 0 | 0 | **3** |
| Wins Acc. | 1 | 0 | 0 | 1 | 0 | 1 | **2** |
| Wins CE | **3** | 0 | 0 | 0 | 0 | 0 | 2 |
| M. rank OpenML Metric | 2.4 | 5.4 | 2.6 | 4.8 | 6.2 | 5 | **1.6** |
| Mean rank Acc. | 2.4 | 5.7 | 4.7 | 4.4 | 5.7 | 3 | **2.1** |
| Mean rank CE | **1.4** | 5.8 | 4.4 | 5.4 | 4.4 | 4.7 | 1.9 |
| Mean Acc. | 0.793±.031 | 0.763±.052 | 0.775±.052 | 0.763±.039 | 0.761±.035 | 0.784±.031 | **0.794**±.033 |
| Mean CE | 0.454±.039 | 0.537±.061 | 0.502±.056 | 0.545±.079 | 0.499±.048 | 0.73±.7 | **0.449**±.05 |
| Mean time (s) | 3182 | 3611 | 3609 | 2877 | 3600 | 3400 | **4.374** (CPU) |

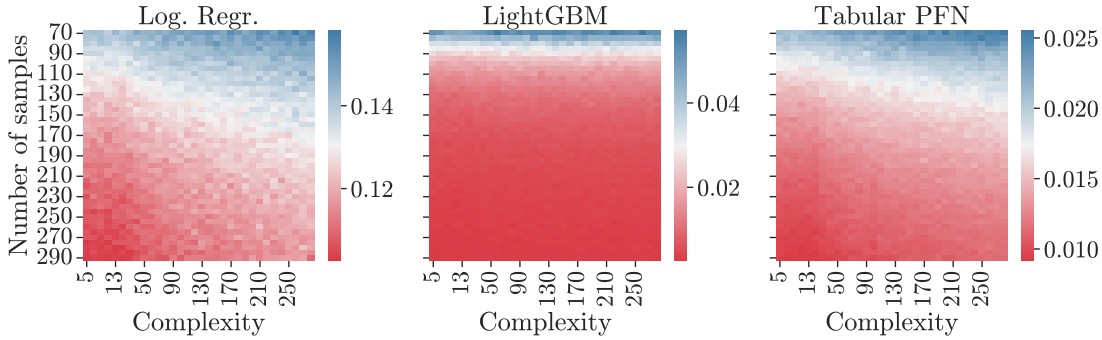

Figure 8: Mean training set uncertainty (cross-entropy loss) on synthetic data generated from random SCMs. The number of training samples is varied on the y-axis and the complexity of the generated data (number of hidden units in the data generating graph) on the x-axis. The cross-entropy mean is averaged across the 100 samples and 1 000 SCMs for each point.

30% of the features, but constantly diminishes. The MLP, which is less not robust to uninformative features, performs even better when the 20% least informative features are removed.

We use our testing tasks from the OpenML CC-18 Benchmark, and, to simplify analyses, we drop multiclass datasets and datasets that contain more than 50 features (as adding 100% more features to a dataset with more than 50 features yields more than 100 features, which TabPFN cannot handle).

### B.3.3 INVARIANCE TO FEATURE ROTATION

Each feature of a tabular dataset typically carries meaning individually, as expressed by column names, such as age or sex. A learning algorithm is rotationally invariant in the sense of Ng (2004), if it is left unchanged when a rotation (unitary) matrix is applied to the features of both the training and testing set, i.e. when features are mixed. To remove uninformative features under a feature rotation, an algorithm first has to restore the original orientation of the features, and then select informative ones. A rotationally invariant algorithm discards the data orientation and thus has to restore the original orientation internally. Thus, Ng (2004) shows that any rotationally invariant learning algorithm has a worst-case sample complexity that grows at least linearly in the number of irrelevant features.

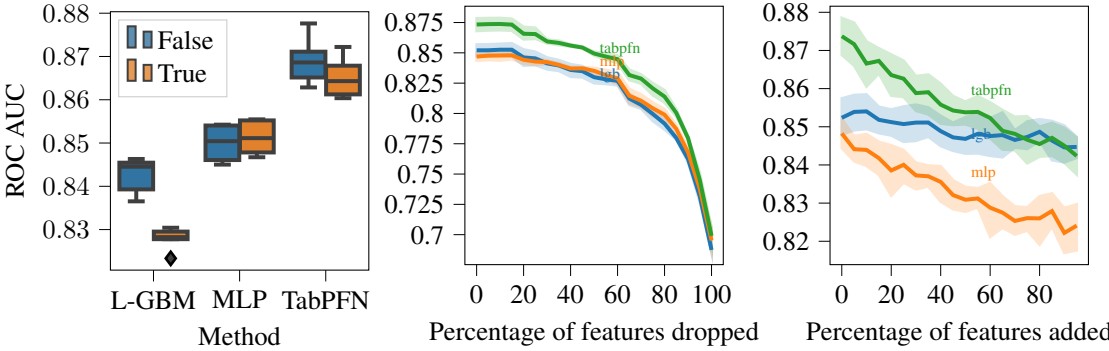

Figure 9: Left: Performance of LightGBM, MLP and TabPFN when random rotations are applied to the feature space. LightGBM loses most predictive accuracy when rotations are applied, MLP is unaffected and TabPFN performs only slightly worse. Center: Iteratively removing features according to their reversed importance rank obtained from a random forest leads to MLPs initially performing better and overtaking LightGBM performance. TabPFN looses predictive accuracy and performs similar to baselines, when most features are removed. Right: Adding uninformative features leads to performance degradation of MLPs and TabPFN, while LightGBM remains relatively constant.

Figure 9 (left) shows the change in test ROC AUC when randomly rotating our datasets, and confirms that only MLPs are rotationally invariant. GBDT methods are highly sensitive to rotations, while TabPFN is less sensitive, but still performs better when no rotations are applied.

The theoretical results by Ng (2004) and empirical results by Grinsztajn et al. (2022) imply, that TabPFN's diminishing performance when uninformative features are added is linked to it's relative rotation invariance. Adjusting the prior to include more uninformative features could address these results.

We use datasets from our testing tasks from the OpenML CC-18 Benchmark with numerical features only, since rotating categorical datasets is problematic, as some GBDT classifiers treat categorical variables distinctly (e.g. generating embeddings per category).

### B.4 ABLATION ON THE SELECTION OF PRIOR MODELS

We perform ablation experiments for a prior based solely on BNNs, SCMs and a mix of both using a hyperparameter to control the sampling likelihood during prior-fitting. The PFNs for each prior are fitted using less compute than in our final experiments, thus their scores generally are slightly worse. The BNN prior, similar to the one used in Müller et al. (2022), provides diminished performance compared to the priors based on SCMs. Additionally, we can see that mixing BNN and SCM prior does not seem to make a big difference on our test set compared to a pure SCM prior.

|  | BNN | SCM | SCM + BNN |
|---|---|---|---|
| Mean CE | 0.811±0.009 | **0.771**±0.006 | 0.776±0.009 |
| Mean ROC AUC | 0.865±0.007 | 0.881±0.002 | **0.883**±0.003 |

Table 4: An evaluation of the impact of the prior mixing on the final performance. Our final model was trained in the *SCM + BNN* setting (see Section 4 for details on the priors).

### B.5 EXTENDED ANALYSIS ON A LARGER BENCHMARK OF DATASETS

We now provide an extended benchmark where we include an analysis on the additional 149 validation datasets[9] (listed in Table 8) in order to assess the generality of our results and to better understand

---

[9]The dataset $flags$ was removed as not enough splits could be generated by our code.

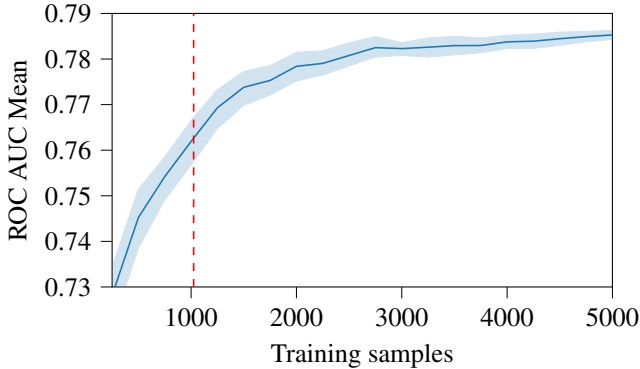

Figure 10: Extrapolation performance of our TabPFN to dataset sizes never seen during training. Maximum number of samples during training was 1024 (dashed red line). The shading indicates the 95% confidence interval over random data splits. A method that does not generalize, would be expected to flatten at 1024.

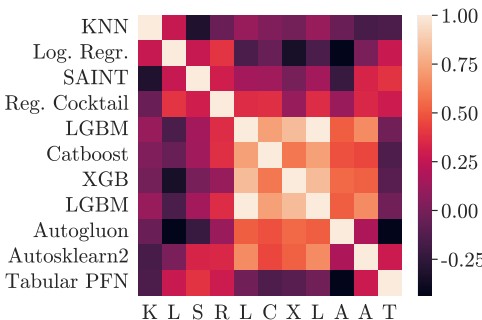

Figure 11: Spearman correlation of per-dataset normalized ROC AUC performance (i.e. the ranking correlation of per-dataset normalized ROC AUC scores) between the considered methods. Ordering on the x-axis is the same as on the y-axis. The TabPFN performs well on a different set of datasets than the baselines, i.e. the Spearman correlation with the GBDT methods is low, while GBDT and AutoML methods are highly correlated, and thus perform well on the same datasets. TabPFN correlates stronger with DL based methods (SAINT and Reg. Cocktail), which, however, do not perform as well as GBDT methods in terms of absolute ROC AUC performance (see 1)

its strengths and weaknesses. We now also include comparisons with default random forests (RFs), support vector machines (SVMs), default XGBoost, etc. , along with their tuned versions after one hour. This evaluation used 5 splits and followed the same experimental setup described in Appendix F.

Figure 12 shows results for our 30 test datasets (separated into the 18 purely numerical datasets without missing values we focus on in the main paper, and the 12 remaining ones), and Figure 13 shows results for the 149 validation datasets (again split into numerical datasets without missing values and the remainder). These figures demonstrate that on the purely numerical datasets without missing values, TabPFN Pareto-dominates all other methods in aggregate results across datasets, in terms of tradeoffs between ROC and time spent. For categorical datasets and datasets with missing values, it still performs well, but not nearly as well as for the numerical case without missing values we targeted.

We now also use statistical tests. We note that when evaluating a new classifier, performance on one dataset is just one data point. In order to apply statistical tests, we need to perform analyses across datasets. The standard method for this are critical difference diagrams as introduced by Demšar

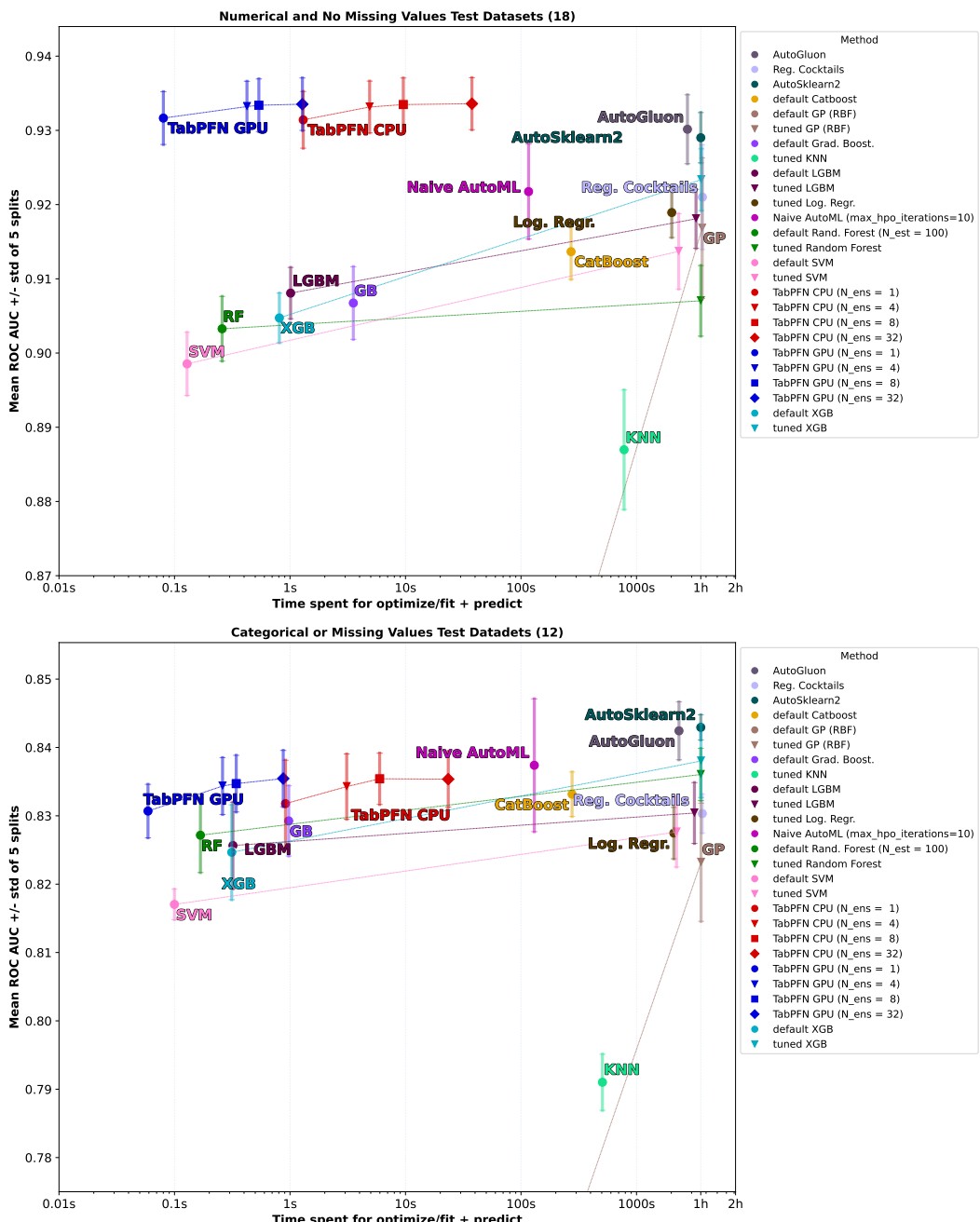

Figure 12: ROC AUC comparison on the OpenML-CC18 Benchmark. Baselines were tuned for one hour or until 10000 configurations were exhausted (Log. Reg and KNN).

(2006). Figure 14 performs these statistical tests to compare methods in two regimes: fast runs that require up to 30 seconds and tuned runs that require up to an hour. We note again that TabPFN is much stronger on purely numerical datasets without missing features. In this case, for the short runs, it statistically significantly outperforms all other baselines. For long runs, it also statistically significantly outperforms all other methods except the AutoML frameworks in 1-vs-1 comparisons; however, this is not visible in the figure because of the multiple testing correction applied due to the multiple pairwise tests.

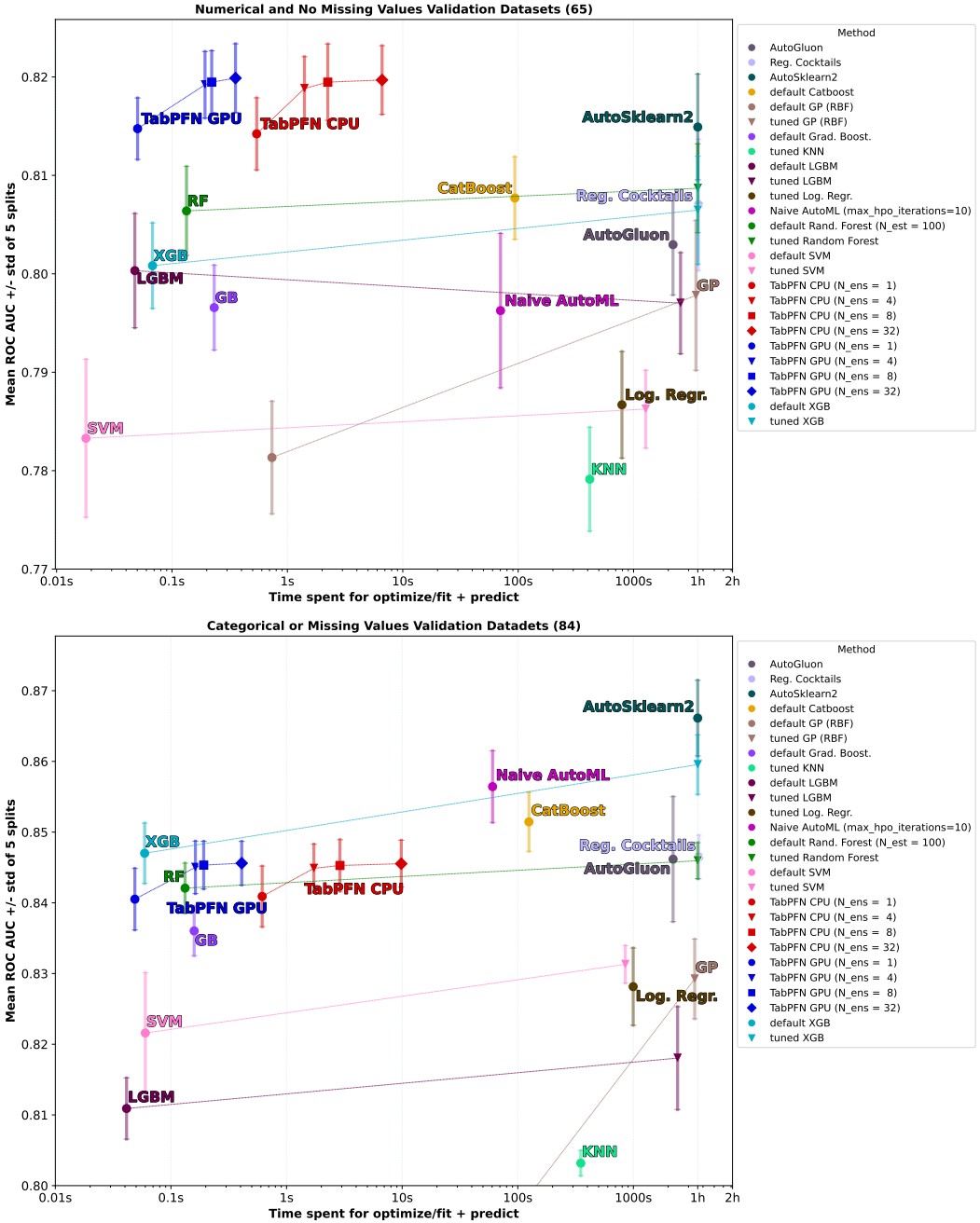

Figure 13: ROC AUC comparison on 149 validation datasets (see Table 8). Baselines were tuned for one hour or until 10 000 configurations were exhausted (Log. Reg and KNN).

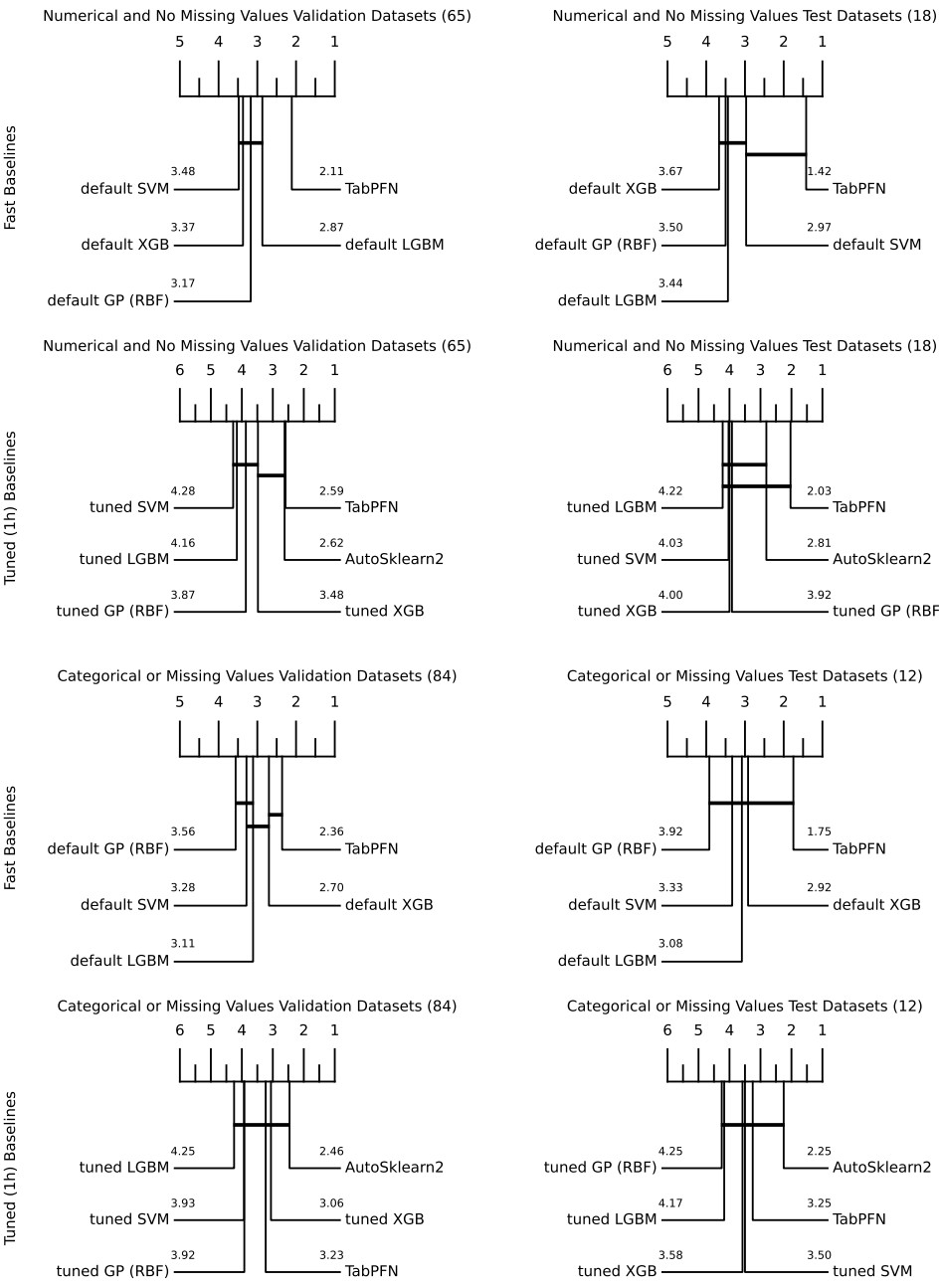

Figure 14: Critical difference plots on average ranks with a Wilcoxon significance analysis. We show plots for both our test set (OpenML-CC18) and our validation set. We split each into a subset of purely numerical datasets without missing values and the rest. We compare to two sets of baselines. i) *Fast Baselines* that finish tuning, training and prediction in less than 30 seconds on average (even the TabPFN on CPU is within this bound). ii) *Tuned Baselines* with one hour of budget for tuning, training and prediction (TabPFN still requires less than 30 seconds on average with the same hardware). We note that in the fast run regime, on numerical datasets without missing values, TabPFN statistically significantly outperforms all other methods.

While the analysis above aggregates across datasets, variation across datasets is large and TabPFN by no means works best on *all* datasets. All classifiers have worst and best cases, and we believe that it is important to complement our reports of TabPFN's strong aggregate performance with the examples we encountered for which the current version does *not* do well. We do show results for all datasets one-by-one at `https://github.com/automl/TabPFNResults/blob/main/individual_plots.pdf`.

The dataset with the worst relative performance for TabPFN is `collins`, with results shown in Figure 15. While most other methods obtain a ROC AUC close to, or identical to 1.0, TabPFN only achieves around 98%. We found this to be caused by uninformative features, which we have already shown to affect TabPFN's performance negatively in Section B.3.2. With just the 5 most important features (as judged by a random forest), TabPFN also achieves an accuracy of 1.0. (In the aggregate analysis above, we of course count the original, poor performance of TabPFN, to not cheat.)

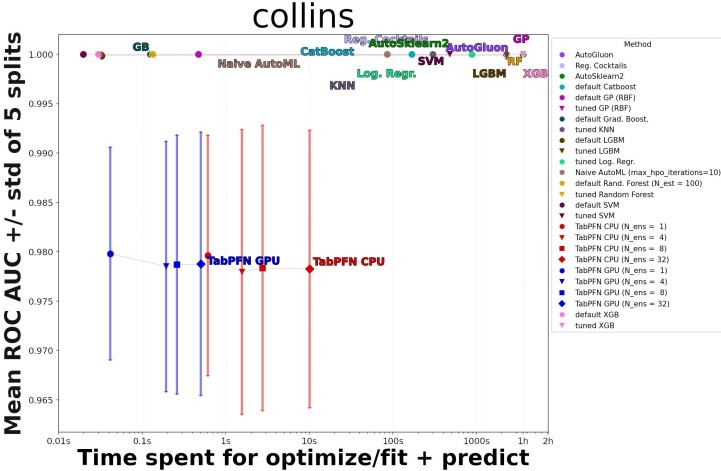

Figure 15: Worst result for TabPFN: `collins`, due to uninformative features. We did not focus on uninformative features in creating TabPFN. With just the 5 most important features (as judged by a random forest), TabPFN also achieves an accuracy of 1.0.

TabPFN also works poorly on purely categorical datasets (e.g., dataset `sensory`, Figure 16a) or with many missing values (e.g., dataset `meta`, Figure 16b).

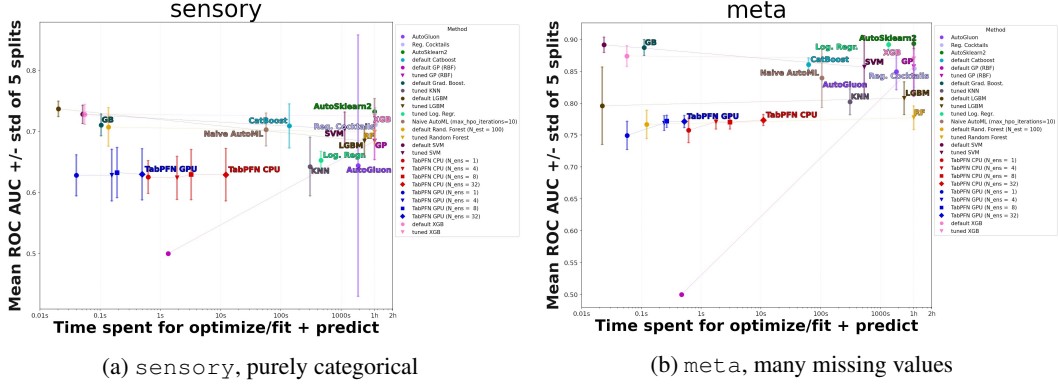

(a) `sensory`, purely categorical

(b) `meta`, many missing values

Figure 16: Further poor results for TabPFN, on datasets with categorical features and missing values, which we did not focus on in creating TabPFN.

TabPFN works best for purely numerical datasets, with no missing values, e.g., datasets `Touch2` and `vehicle`, see Figure 17.

TabPFN still works well in several datasets where hyperparameter optimization does not help in the baselines (maybe due to overfitting), e.g. `Pizza-cutter1` and `arsenic-female-bladder`,

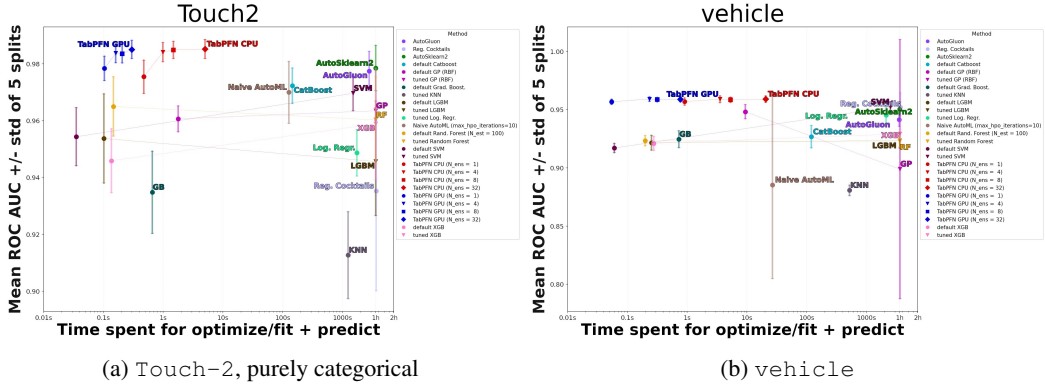

(a) `Touch-2`, purely categorical    (b) `vehicle`

Figure 17: Strong results for TabPFN for individual datasets with numerical features and no missing values.

see Figure 18. We attribute this to being Bayesian, having meta-learned not to overfit on small datasets with a simplicity prior using principles from causality.

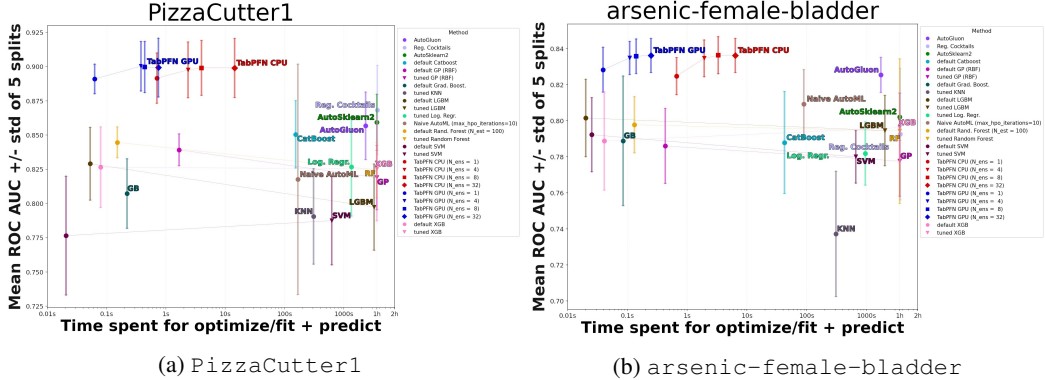

(a) `PizzaCutter1`    (b) `arsenic-female-bladder`

Figure 18: TabPFN can still yield strong results when hyperparameter optimization does not help (potentially due to overfitting).

It is *not* the case, however, that TabPFN works great on all numerical datasets without missing values and poorly on all categorical datasets. For example, TabPFN works poorly on the numerical dataset `pm10` and great on the categorical dataset `monks-problem2`, see Figure 19. It is therefore necessary to look beyond individual datasets to obtain the entire picture.

## C    DETAILS OF THE TABPFN PRIOR

### C.1    SCM PRIOR

**The Sampling Algorithm**    We instantiate a subfamily of DAGs that can be efficiently sampled from by starting with a MLP architecture and dropping weights from it. That is, to sample a dataset with $k$ features and $n$ samples from our prior we perform the following steps for each dataset:

(1) We sample the number of MLP layers $l \sim p(l)$ and nodes $h \sim p(h)$ and sample a graph $\mathcal{G}(Z, E)$ structured like an $l$-layered MLP with hidden size $h$.

(2) We sample weights for each Edge $E_{ij}$ as $W_{i,j} \sim p_w(\cdot)$.

(3) We drop a random set of edges $e \in E$ to yield a random DAG.

(4) We sample a set of $k$ feature nodes $N_x$ and a label node $N_y$ from the nodes $Z$.

(5) We sample the noise distributions $p(\epsilon) \sim p(p(\epsilon))$ from a meta-distribution. This yields an SCM, with all $f_i$'s instantiated as random affine mappings followed by an activation. Each $z_i$ corresponds to a sparsely connected neuron in the MLP.

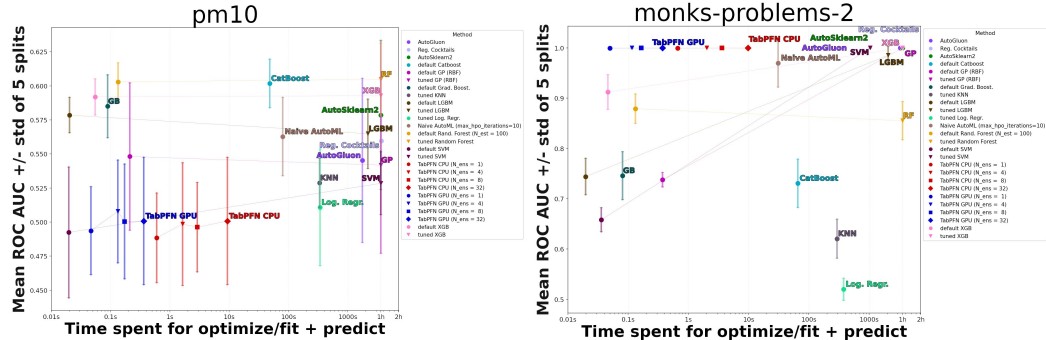

(a) `pm10`, purely numerical, but still poor performance (b) `monks-problem2`, categorical, but still strong performance

Figure 19: TabPFN can also yield poor performance on numerical datasets (`pm`) and strong performance on categorical datasets (`monks-problem2`.

With the above parameters fixed, we perform the following steps for each member of the dataset:
(1) We sample noise variables $\epsilon_i$ from their specific distributions.
(2) We compute the value of all $z \in Z$ with $z_i = a((\sum_{j \in \text{PA}_{\mathcal{G}(i)}} E_{ij} z_j) + \epsilon_i)$.
(3) We retrieve the values at the feature nodes $N_x$ and the output node $N_y$ and return them.

We sample one activation function $a$ per dataset from $\{Tanh, LeakyReLU, ELU, Identity\}$ (Nair and Hinton, 2010). The sampling scheme for the number of layers $p(l)$ and nodes $p(h)$ is designed to follow a discretized noisy log-normal distribution, $p(\epsilon)$ is a noisy log-normal distribution and the dropout rate follows a beta distribution. The full information can be found in Table 5.

## C.2 TABULAR DATA REFINEMENTS

Tabular datasets comprise a range of peculiarities, e.g. feature types can be numerical, ordinal, or categorical and feature values can be missing, leading to sparse features. We seek to reflect these peculiarities in the design of our prior as described in the following sections.

### C.2.1 PREPROCESSING

During prior-fitting, input data is normalized to zero mean and unit variance, and we apply the same step when evaluating on real data. Since tabular data frequently contains exponentially scaled data, which might not be present during prior-fitting we apply power scaling during inference (Yeo and Johnson, 2000). Thus, during inference on real tabular datasets the features more closely match those seen during prior-fitting. We use only training samples for calculating z-statistics, power transforms and all other preprocessing. We take this preprocessing time into account when reporting the inference time of our method.

### C.2.2 CORRELATED FEATURES

Feature correlation in tabular data varies between datasets and ranges from independent to highly correlated. This poses problems to classical deep learning methods (Borisov et al., 2021). When considering a large space of SCMs, correlated features of varying degrees naturally arise in our priors. Furthermore, in real-world tabular data, the ordering of features is often unstructured, however adjacent features are often more highly correlated than others. We use "Blockwise feature sampling" to reflect the correlation structure between ordered features. Our generation method of SCMs naturally provides a way to do this. The first step in generating our SCMs is generating a unidirectional layered network structure in which nodes in one layer can only receive inputs from the preceding layer. Thus, features in the same layer tend to be more highly correlated. We use this by sampling adjacent nodes in the layered network structure in blocks and using these ordered blocks in our set of features. In Figure 20, we visualize the correlations of such a generated dataset (right) and compare them to a

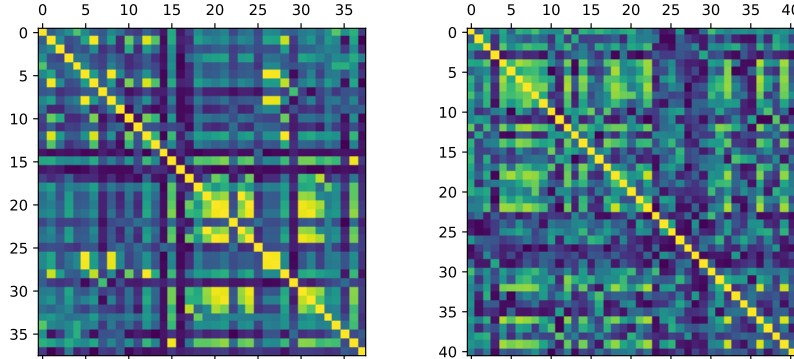

Figure 20: Feature correlation matrices for a real-world ("PC4 Software defect prediction", left) and a synthetic (right) dataset, where brighter colors indicate higher correlation.

real-world dataset (left), demonstrating that our prior yields correlation structures similar to those of real datasets.

### C.2.3 GENERATING IRREGULAR FUNCTIONS

In real-world data, some features are consistently more important than others. While a random network weight initialization leads to slightly different feature importances, the average effect of input features regresses to the mean when the hidden dimensionality increases. We amplify differences by sampling a weight parameter for each input feature and multiplying all outgoing weights by this factor. In the prior, we randomly sparsify connections of the graph. Thus hidden variables and the output node are influenced by fewer parameters, yielding more irregular patterns, as a larger number of parameters once again regresses to the mean. We also extend sparsification to blocks of variables, leading to some groups of variables interacting more strongly. We also extend the way noise variables are sampled. Instead of sampling Gaussian noise at each node from the same distribution, we first sample separate noise means and standard deviations for each node and then sample from this distribution. Also, we generate non-uniformly distributed input data x, as observed in real-world data: We sample the input variables x (which are propagated through our network), from a mix of distributions, namely the Gaussian, Zipfian and Multivariate Distribution.

### C.2.4 NAN HANDLING

We do not have special nan handling built into our model. We replace nan values with zero at test time.

### C.2.5 CATEGORICAL FEATURES

Tabular data often includes not only numeric features but also discrete categorical ones. While categorical features should technically not be ordered, in practice, they sometimes are, i.e., the categories represent binned degrees of some underlying variable. We introduce categorical features by picking a random fraction $p_{cat}$ (a hyperparameter) of categorical features per dataset. Analogous to transforming numeric class labels to discrete multiclass labels, we convert dense features to discrete ones. Also analogous to multiclass labels, we pick a shuffling fraction of categorical features $p_{scat}$ where we reshuffle categories. For details, see Section 4.5. During prior-fitting, we use a probability for categorical features of 20%.

### C.2.6 DIFFERENCES TO PRIOR WORK ON PFNs FOR TABULAR DATA

Prior work by Müller et al. (2022) has demonstrated tabular data classification using PFNs, but was limited to 30 training samples, balanced binary-classification and 60 features. Here we summarize the most important changes, to this prior work:

i) The PFNs for tabular data described by Müller et al. (2022) can only handle balanced binary datasets. In Section 4.5, we show how to extend the prior to handle imbalanced classes and multi-class classification problems.

ii) We studied pre-processing techniques for the TabPFN, which was not done at all before. This includes outlier removal and power scaling during inference (Yeo and Johnson, 2000). During training, we also rotate feature indices (based on the assumptions that it should not be informative whether a feature is listed in the $i$-th or the $j$-th column, but that features may be listed in groups and that there may be more interactions between features with similar indices[10]) and class labels (based on the same assumptions).

iii) We construct ensembles over multiple pre-processings. Specifically, ensemble members differ in the feature column rotation, the class label rotation, and the use of a power transform. We only include each combination of the $k$ possible feature column rotations (for $k$ features), $j$ possible class label rotations (for $j$-ary classification problems) and 2 choices of transformation (power transform, none). Even if a number of ensemble members is specified that is larger than $2kj$, we only use $2kj$ ensemble members.

iv) We changed the transformer architecture to make it faster. We shrank attention matrix sizes from $(n + m)^2$ to $n^2 + n * m$, for $n$ training points and m inference points.

v) We introduce a novel SCM prior. We compare the SCM, the improved BNN prior (which is more heavily based on the BNN setup of C.2.6) and SCM + BNN in Table 4. This improvesperformance by 2% in this smaller scale setup, which is a bigger difference than between the performance of the final TabPFN and all baselines besides KNN and SAINT.

## D   DETAILS OF THE PRIOR-DATA FITTED NETWORK ALGORITHM

Algorithm 1 describes the training method proposed by Müller et al. (2022) for PFNs.

---

**Algorithm 1:** Prior-fitting of a PFN (Müller et al., 2022)

---

**Input** : A prior distribution over datasets $p(D)$, from which samples can be drawn and the number of samples $K$ to draw
**Output :** A model $q_\theta$ that will approximate the PPD
Initialize the neural network $q_\theta$;
**for** $j \leftarrow 1$ **to** $K$ **do**
    Sample $D \cup \{(x_i, y_i)\}_{i=1}^{m} \sim p(D)$;
    Compute stochastic loss approximation $\bar{\ell}_\theta = \sum_{i=1}^{m} (- \log q_\theta(y_i | x_i, D))$;
    Update parameters $\theta$ with stochastic gradient descent on $\nabla_\theta \bar{\ell}_\theta$;
**end**

---

## E   SETUP OF OUR METHOD

### E.1   TRANSFORMER HYPERPARAMETERS

We considered only PFN Transformers with 12 layers, embeddings size 512, hidden size 1024 in feed-forward layers, and 4-head attention. We used the Adam optimizer (Kingma and Ba, 2015) with linear-warmup and cosine annealing (Loshchilov and Hutter, 2017). For each training we tested a set of 3 learning rates, $\{.001, .0003, .0001\}$, and used the one with the lowest final training loss. The resulting model contains 25.82 M parameters.

### E.2   PFN ARCHITECTURE ADAPTATIONS

**Attention Adaption** The original PFN architecture (Müller et al., 2022) uses a single multi-head self-attention module (Vaswani et al., 2017) to compute the attention between all the training examples, as well as, the attention from validation examples to training examples. We replaced this, with two

---

[10]To avoid misunderstandings and give an example, a rotation of column indices by 2 positions would change the columns of the $X$ matrix from $[x_1, x_2, x_3, x_4]$ to $[x_3, x_4, x_1, x_2]$. This is *not* a rotation of the $X$ matrix, just of its column indices.

modules that share weights, one which computes self-attention among the training examples and the other that only compute cross-attention from validation examples to training examples. Conceptually, this is equivalent to the original architecture, except that we're using a slightly different self-attention mask than the original architecture, which allowed all examples to attend to itself (the diagonal is 1), as in this example:

$$\begin{bmatrix} 1 & 1 & 1 & 0 & 0 \\ 1 & 1 & 1 & 0 & 0 \\ 1 & 1 & 1 & 0 & 0 \\ 1 & 1 & 1 & 1 & 0 \\ 1 & 1 & 1 & 0 & 1 \end{bmatrix}. \tag{3}$$

For validation examples, we remove the attention to themselves. In terms of the example above:

$$\begin{bmatrix} 1 & 1 & 1 & 0 & 0 \\ 1 & 1 & 1 & 0 & 0 \\ 1 & 1 & 1 & 0 & 0 \\ 1 & 1 & 1 & 0 & 0 \\ 1 & 1 & 1 & 0 & 0 \end{bmatrix}. \tag{4}$$

Information about the state of the current position does still flow through the residual branch, though.

**Flexible Encoder** Datasets have unequal numbers of input dimensions (features), while PFNs use an encoder layer that accepts fixed dimensional inputs. Here we explain how datasets with different numbers of dimensions can be modelled with a single PFN: We draw the number of dimensions of a dataset during training uniformly at random up to 100. Our encoder changes to accomodate this training and inference with different numbers of features by zero-padding datasets where the number of features $k$ is smaller than the maximum number of features $K$ and scaling these features by $\frac{K}{k}$, s.t. the magnitude stays the same.

### E.3 TabPFN Training

We trained our final model for $18\,000$ steps with a batch size of $512$ datasets. That is our TabPFN is trained on $9\,216\,000$ synthetically generated datasets. This training takes 20 hours on 8 GPUs (Nvidia RTX 2080 Ti). Each dataset had a fixed size of $1024$ and we split it into training and validation uniformly at random. We generally saw that learning curves tended to flatten after around 10 million datasets and were generally very noisy. Likely, this is because our prior generates a wide variety of different datasets.

### E.4 Prior Hyperparameters

The hyperparameters of our prior were chosen based on simplicity and our observations on the validation datasets (such as their class distributions or feature correlation strengths). Also, during algorithm development, we evaluated our models on this set of datasets to decide if our developed methods were correct and working. Since our prior hyperparameters specify distributions and not definite values, they can be chosen over a wide range and resemble the intervals chosen for a random hyperparameter search. The prior distributions we used are given in Table 5.

## F Details for Tabular Experiments

Here we provide additional details for the experiments conducted in Section 5 in the main paper.

### F.1 Hardware Setup

All evaluations, including the baselines, ran on a compute cluster equipped with Intel(R) Xeon(R) Gold 6242 CPU @ 2.80GHz using 1 CPU with up to 6GB RAM. For evaluation using our TabPFN, we additionally use an RTX 2080 Ti.

Table 5: Overview of our prior hyperparameter distribution. For many features we use a Log Uniform distribution with truncated normal noise, which we refer to as $\text{TNLU}(h|\check{\mu}, \hat{\mu}, min, round)$. We sample from it by first sampling mean $\mu$ and standard deviation $\sigma$ from $\mu, \sigma \sim \text{LogUniform}(\check{\mu}, \hat{\mu})$ and then sampling from the resulting truncated normal distribution $v \sim \text{TruncNormal}(\mu, \sigma^2, a = 0, b = inf)$. $v$ is rounded to the closest integer, if $round$ is set. The final sampled value then is $h = v + min$.

| | Sampling distribution $p(\psi)$ | | | | |
|---|---|---|---|---|---|
| MLP weight dropout | $0.9 \cdot \text{Beta}(a, b)$, where $a, b \sim \text{Uniform}(0.1, 5.0)$ | | | | |
| | Choices | | | | |
| Sample SCM vs BNN | Uniform Choice | {True, False} | | | |
| Share Noise mean for nodes | Uniform Choice | {True, False} | | | |
| Input feature scaling enabled | Uniform Choice | {True, False} | | | |
| Sample y from last MLP layer | Uniform Choice | {True, False} | | | |
| MLP Activation Functions | Uniform Choice | {Tanh, Leaky ReLU, ELU, Identity} | | | |
| Blockwise Dropout | Uniform Choice | {True, False} | | | |
| Keep SCM feature order | Uniform Choice | {True, False} | | | |
| Sample feature nodes blockwise | Uniform Choice | {True, False} | | | |
| | | Max Mean $\hat{\mu}$ | Min Mean $\check{\mu}$ | $round$ | $min$ |
| MLP #layers | TNLU | 6 | 1 | True | 2 |
| MLP #hidden nodes per layer | TNLU | 130 | 5 | True | 4 |
| Gaussian Noise Std. | TNLU | 0.3 | 0.0001 | False | 0.0 |
| MLP Weights Std. | TNLU | 10.0 | 0.01 | False | 0.0 |
| SCM #nodes at layer 1 | TNLU | 12 | 1 | True | 1 |

## F.2 BASELINES

We provide the search space used to tune our baselines in Table 6. For *CatBoost* and *XGBoost*, we used the same ranges as Shwartz-Ziv and Armon (2022) with the following exception: For *CatBoost* we removed the hyperparameter max_size since we could not find it in the official documentation. To be maximally fair to XGBoost, we also tried the search space of quadruple Kaggle grandmaster Bojan Tunguz (Tunguz, 2022), which we adapted slightly by using softmax instead of logistic, as we are in the multi-class setting. XGBoost with this search space performed worse for all considered time budgets than the search space by Shwartz-Ziv and Armon (2022). The search spaces for the KNN, GP and Logistic Regression baselines were designed from scratch and we used the respective implementation from *scikit-learn* (Pedregosa et al., 2011). For *CatBoost* and *AutoSklearn*, we pass the position of categorical features to the classifier (*AutoGluon* automatically detects categorical feature columns). We normalize inputs for Logistic Regression, GP and KNN to the range [0, 1] using MinMax Scaling.

## F.3 USED DATASETS

To construct and evaluate our method, we used the following four sets of datasets.

First, our meta-test set (see Table 7) comprises all datasets in the OpenML-CC18 benchmark suite (Bischl et al., 2021)(available at `OpenML.org`) with at most 2 000 samples, 100 features and 10 classes, which leaves us with 30 datasets that represent small, tabular datasets.

Second, our meta-validation set (see Tables 8 and 9) comprises 150 datasets from `OpenML.org` (Vanschoren et al., 2014). For this, we considered all datasets on `OpenML.org` and applied the following filtering procedure: We dropped all datasets that are in the meta-test set and all datasets with more than 2 000 samples, 100 features or 10 classes. We also manually checked for overlaps and removed datasets where the number of features, classes and samples was identical to a dataset in the meta-test set. Furthermore, we manually dropped FOREX (since it is a time series dataset) and artificially-created datasets, such as the Univ and Friedman datasets. The remaining meta-validation set then contains 150 datasets. This meta-validation set was used to guide the development of our prior hyperparameters as described in Appendix E.4.

Table 6: Hyperparameter spaces for baselines. All, except LightGBM, adapted from Shwartz-Ziv and Armon (2022).

| baseline | name | type | log | range |
|---|---|---|---|---|
| LogReg | penalty | cat | (l1, l2, none) | - |
| | max_iter | int | [50, 500] | - |
| | fit_intercept | cat | (True, False) | - |
| | C | float | $[e^{-5}, 5]$ | - |
| KNN | n_neighbors | int | [1, 16] | - |
| GP | params_y_scale | float | [0.05, 5.0] | yes |
| | params_length_scale | float | [0.1, 1.0] | yes |
| CatBoost | learning_rate | float | $[e^{-5}, 1]$ | yes |
| | random_strength | int | [1, 20] | - |
| | l2_leaf_reg | float | [1, 10] | yes |
| | bagging_temperature | float | [0, 1.0] | yes |
| | leaf_estimation_iterations | int | [1, 20] | - |
| | iterations | int | [100, 4000] | - |
| XGBoost | learning_rate | float | $[e^{-7}, 1]$ | yes |
| | max_depth | int | [1, 10] | - |
| | subsample | float | [0.2, 1] | - |
| | colsample_bytree | float | [0.2, 1] | - |
| | colsample_bylevel | float | [0.2, 1] | - |
| | min_child_weight | float | $[e^{-16}, e^5]$ | yes |
| | alpha | float | $[e^{-16}, e^2]$ | yes |
| | lambda | float | $[e^{-16}, e^2]$ | yes |
| | gamma | float | $[e^{-16}, e^2]$ | yes |
| | n_estimators | int | [100, 4000] | - |
| LightGBM | num_leaves | int | [5, 50] | yes |
| | max_depth | int | [3, 20] | yes |
| | learning_rate | float | $[e^{-3}, 1]$ | - |
| | n_estimators | int | 50, 2000 | - |
| | min_child_weight | float | $[e^{-5}, e^4]$ | yes |
| | reg_alpha | float | [0, 1e-1, 1, 2, 5, 7, 10, 50, 100] | yes |
| | reg_lambda | float | [0, 1e-1, 1, 5, 10, 20, 50, 100] | yes |
| | subsample | float | [0.2, 0.8] | - |

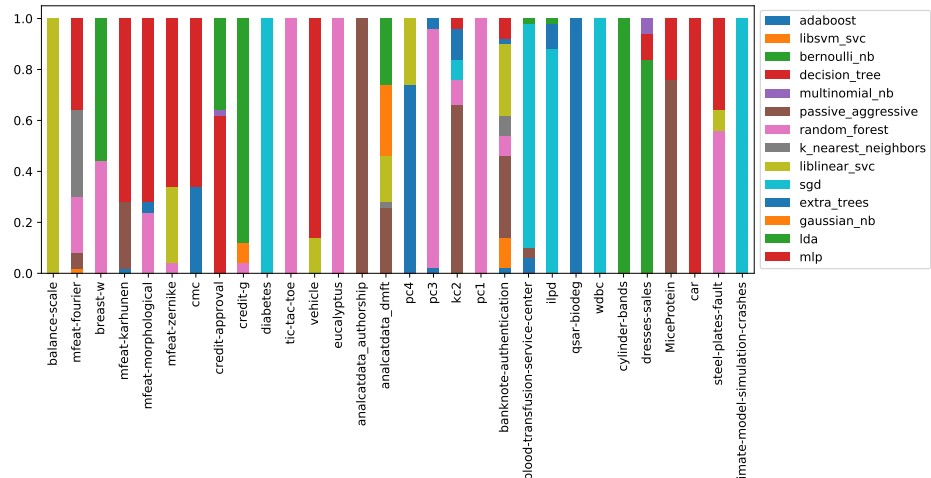

Figure 21: Ensemble weights of classifiers used in AutoSklearn baseline for each dataset. Ensemble weights are averaged across 5 splits for each dataset in the OpenML-CC18 Benchmark after one hour of training.

Third, a subset of 5 datasets from the OpenML-AutoML Benchmark, comprises all datasets from the OpenML-AutoML Benchmark with at most 1 111 samples, 100 features and 10 classes. This is given in Table 10. Due to the 10-fold cross-validation in the OpenML-AutoML Benchmark this is identical to the setup for our meta-test and meta-validation datasets above, where we used up to 2 000 samples split 50-50 into training and test.

Fourth, our meta-generalization set, described in Appendix F.4, comprises 18 larger datasets from the OpenML AutoML Benchmark.

## F.4   MODEL GENERALIZATION

For testing the TabPFN performance on longer sequences, as described in Section 10, we used a set of 18 datasets from the OpenML AutoML Benchmark that contain at least 10 000 samples. The list of datasets used can be found in Table 11. For this evaluation, datasets with more than 100 features are limited to the first 100 features. When more than 10 classes are contained in the datasets, samples with any but the first 10 classes are discarded.

## F.5   DETAILS ON TIME COMPARISONS

Time comparisons refer to combined fitting, tuning and prediction; see Table 2 for the times split into tuning/fitting and prediction. The time taken for each baseline and TabPFN does not include the one-time cost of development of each method. Thus, for AutoML baselines meta-learning cost was not included (e.g. Auto-Sklearn pipelines meta-learning, which involved running hyper-parameter search for 24 hours on 140 datasets (= 3360 CPU hours) (Feurer et al., 2021)). For GBDT methods the manual time that went into defining suitable hyperparameter spaces and the manual crafting of algorithms that perform well on tabular datasets is hard to measure and was not included. For TabPFN the prior-fitting phase (which is part of our algorithm development: i.e. developing ideas, writing code, trying ideas) is not included. This is fair to all methods, as these costs are not on the user side and are amortized over time.

Table 7: Datasets used for the evaluation. These include all 30 datasets in the OpenML-CC18 benchmark suite with at most 2 000 samples, 100 features and 10 classes.

| Name | #Feat. | #Cat. | #Inst. | #Class. | #NaNs | Minor. Class Size | OpenML ID |
|---|---|---|---|---|---|---|---|
| balance-scale | 5 | 1 | 625 | 3 | 0 | 49 | 11 |
| mfeat-fourier | 77 | 1 | 2000 | 10 | 0 | 200 | 14 |
| breast-w | 10 | 1 | 699 | 2 | 16 | 241 | 15 |
| mfeat-karhunen | 65 | 1 | 2000 | 10 | 0 | 200 | 16 |
| mfeat-morphological | 7 | 1 | 2000 | 10 | 0 | 200 | 18 |
| mfeat-zernike | 48 | 1 | 2000 | 10 | 0 | 200 | 22 |
| cmc | 10 | 8 | 1473 | 3 | 0 | 333 | 23 |
| credit-approval | 16 | 10 | 690 | 2 | 67 | 307 | 29 |
| credit-g | 21 | 14 | 1000 | 2 | 0 | 300 | 31 |
| diabetes | 9 | 1 | 768 | 2 | 0 | 268 | 37 |
| tic-tac-toe | 10 | 10 | 958 | 2 | 0 | 332 | 50 |
| vehicle | 19 | 1 | 846 | 4 | 0 | 199 | 54 |
| eucalyptus | 20 | 6 | 736 | 5 | 448 | 105 | 188 |
| analcatdata_auth... | 71 | 1 | 841 | 4 | 0 | 55 | 458 |
| analcatdata_dmft | 5 | 5 | 797 | 6 | 0 | 123 | 469 |
| pc4 | 38 | 1 | 1458 | 2 | 0 | 178 | 1049 |
| pc3 | 38 | 1 | 1563 | 2 | 0 | 160 | 1050 |
| kc2 | 22 | 1 | 522 | 2 | 0 | 107 | 1063 |
| pc1 | 22 | 1 | 1109 | 2 | 0 | 77 | 1068 |
| banknote-authenti... | 5 | 1 | 1372 | 2 | 0 | 610 | 1462 |
| blood-transfusion-... | 5 | 1 | 748 | 2 | 0 | 178 | 1464 |
| ilpd | 11 | 2 | 583 | 2 | 0 | 167 | 1480 |
| qsar-biodeg | 42 | 1 | 1055 | 2 | 0 | 356 | 1494 |
| wdbc | 31 | 1 | 569 | 2 | 0 | 212 | 1510 |
| cylinder-bands | 40 | 22 | 540 | 2 | 999 | 228 | 6332 |
| dresses-sales | 13 | 12 | 500 | 2 | 835 | 210 | 23381 |
| MiceProtein | 82 | 5 | 1080 | 8 | 1396 | 105 | 40966 |
| car | 7 | 7 | 1728 | 4 | 0 | 65 | 40975 |
| steel-plates-fault | 28 | 1 | 1941 | 7 | 0 | 55 | 40982 |
| climate-model-simu... | 21 | 1 | 540 | 2 | 0 | 46 | 40994 |

Table 8: Meta-Datasets used for developing the prior.

| Name | #Feat. | #Cat. | #Inst. | #Class. | #NaNs | Minor. Class Size | OpenML ID |
|---|---|---|---|---|---|---|---|
| breast-cancer | 10 | 10 | 286 | 2 | 9 | 85 | 13 |
| colic | 27 | 20 | 368 | 2 | 1927 | 136 | 25 |
| dermatology | 35 | 34 | 366 | 6 | 8 | 20 | 35 |
| sonar | 61 | 1 | 208 | 2 | 0 | 97 | 40 |
| glass | 10 | 1 | 214 | 6 | 0 | 9 | 41 |
| haberman | 4 | 2 | 306 | 2 | 0 | 81 | 43 |
| tae | 6 | 3 | 151 | 3 | 0 | 49 | 48 |
| heart-c | 14 | 8 | 303 | 2 | 7 | 138 | 49 |
| heart-h | 14 | 8 | 294 | 2 | 782 | 106 | 51 |
| heart-statlog | 14 | 1 | 270 | 2 | 0 | 120 | 53 |
| hepatitis | 20 | 14 | 155 | 2 | 167 | 32 | 55 |
| vote | 17 | 17 | 435 | 2 | 392 | 168 | 56 |
| ionosphere | 35 | 1 | 351 | 2 | 0 | 126 | 59 |
| iris | 5 | 1 | 150 | 3 | 0 | 50 | 61 |
| wine | 14 | 1 | 178 | 3 | 0 | 48 | 187 |
| flags | 29 | 27 | 194 | 8 | 0 | 4 | 285 |
| hayes-roth | 5 | 1 | 160 | 3 | 0 | 31 | 329 |
| monks-problems-1 | 7 | 7 | 556 | 2 | 0 | 278 | 333 |
| monks-problems-2 | 7 | 7 | 601 | 2 | 0 | 206 | 334 |
| monks-problems-3 | 7 | 7 | 554 | 2 | 0 | 266 | 335 |
| SPECT | 23 | 23 | 267 | 2 | 0 | 55 | 336 |
| SPECTF | 45 | 1 | 349 | 2 | 0 | 95 | 337 |
| grub-damage | 9 | 7 | 155 | 4 | 0 | 19 | 338 |
| synthetic_control | 61 | 1 | 600 | 6 | 0 | 100 | 377 |
| prnn_crabs | 8 | 2 | 200 | 2 | 0 | 100 | 446 |
| analcatdata_lawsuit | 5 | 2 | 264 | 2 | 0 | 19 | 450 |
| irish | 6 | 4 | 500 | 2 | 32 | 222 | 451 |
| analcatdata_broadwaymult | 8 | 5 | 285 | 7 | 27 | 21 | 452 |
| analcatdata_reviewer | 8 | 8 | 379 | 4 | 1418 | 54 | 460 |
| backache | 32 | 27 | 180 | 2 | 0 | 25 | 463 |
| prnn_synth | 3 | 1 | 250 | 2 | 0 | 125 | 464 |
| schizo | 15 | 3 | 340 | 2 | 834 | 163 | 466 |
| profb | 10 | 5 | 672 | 2 | 1200 | 224 | 470 |
| analcatdata_germangss | 6 | 5 | 400 | 4 | 0 | 100 | 475 |
| biomed | 9 | 2 | 209 | 2 | 15 | 75 | 481 |
| rmftsa_sleepdata | 3 | 1 | 1024 | 4 | 0 | 94 | 679 |
| diggle_table_a2 | 9 | 1 | 310 | 9 | 0 | 18 | 694 |
| rmftsa_ladata | 11 | 1 | 508 | 2 | 0 | 222 | 717 |
| pwLinear | 11 | 1 | 200 | 2 | 0 | 97 | 721 |
| analcatdata_vineyard | 4 | 2 | 468 | 2 | 0 | 208 | 724 |
| machine_cpu | 7 | 1 | 209 | 2 | 0 | 56 | 733 |
| pharynx | 11 | 10 | 195 | 2 | 2 | 74 | 738 |
| auto_price | 16 | 2 | 159 | 2 | 0 | 54 | 745 |
| servo | 5 | 5 | 167 | 2 | 0 | 38 | 747 |
| analcatdata_wildcat | 6 | 3 | 163 | 2 | 0 | 47 | 748 |
| pm10 | 8 | 1 | 500 | 2 | 0 | 246 | 750 |
| wisconsin | 33 | 1 | 194 | 2 | 0 | 90 | 753 |
| autoPrice | 16 | 1 | 159 | 2 | 0 | 54 | 756 |
| meta | 22 | 3 | 528 | 2 | 504 | 54 | 757 |
| analcatdata_apnea3 | 4 | 3 | 450 | 2 | 0 | 55 | 764 |
| analcatdata_apnea2 | 4 | 3 | 475 | 2 | 0 | 64 | 765 |
| analcatdata_apnea1 | 4 | 3 | 475 | 2 | 0 | 61 | 767 |
| disclosure_x_bias | 4 | 1 | 662 | 2 | 0 | 317 | 774 |
| bodyfat | 15 | 1 | 252 | 2 | 0 | 124 | 778 |
| cleveland | 14 | 8 | 303 | 2 | 6 | 139 | 786 |
| triazines | 61 | 1 | 186 | 2 | 0 | 77 | 788 |
| disclosure_x_tampered | 4 | 1 | 662 | 2 | 0 | 327 | 795 |
| cpu | 8 | 2 | 209 | 2 | 0 | 53 | 796 |
| cholesterol | 14 | 8 | 303 | 2 | 6 | 137 | 798 |
| chscase_funds | 3 | 1 | 185 | 2 | 0 | 87 | 801 |
| pbcseq | 19 | 7 | 1945 | 2 | 1133 | 972 | 802 |
| pbc | 19 | 9 | 418 | 2 | 1239 | 188 | 810 |
| rmftsa_ctoarrivals | 3 | 2 | 264 | 2 | 0 | 101 | 811 |
| chscase_vine2 | 3 | 1 | 468 | 2 | 0 | 212 | 814 |
| chatfield_4 | 13 | 1 | 235 | 2 | 0 | 93 | 820 |
| boston_corrected | 21 | 4 | 506 | 2 | 0 | 223 | 825 |
| sensory | 12 | 12 | 576 | 2 | 0 | 239 | 826 |
| disclosure_x_noise | 4 | 1 | 662 | 2 | 0 | 329 | 827 |
| autoMpg | 8 | 4 | 398 | 2 | 6 | 189 | 831 |
| kdd_el_nino-small | 9 | 3 | 782 | 2 | 466 | 274 | 839 |
| autoHorse | 26 | 9 | 205 | 2 | 57 | 83 | 840 |
| stock | 10 | 1 | 950 | 2 | 0 | 462 | 841 |
| breastTumor | 10 | 9 | 286 | 2 | 9 | 120 | 844 |
| analcatdata_gsssexsurvey | 10 | 6 | 159 | 2 | 6 | 35 | 852 |
| boston | 14 | 2 | 506 | 2 | 0 | 209 | 853 |
| fishcatch | 8 | 3 | 158 | 2 | 87 | 63 | 854 |
| vinnie | 3 | 1 | 380 | 2 | 0 | 185 | 860 |
| mu284 | 11 | 1 | 284 | 2 | 0 | 142 | 880 |
| no2 | 8 | 1 | 500 | 2 | 0 | 249 | 886 |
| chscase_geyser1 | 3 | 1 | 222 | 2 | 0 | 88 | 895 |
| chscase_census6 | 7 | 1 | 400 | 2 | 0 | 165 | 900 |
| chscase_census5 | 8 | 1 | 400 | 2 | 0 | 193 | 906 |
| chscase_census4 | 8 | 1 | 400 | 2 | 0 | 194 | 907 |
| chscase_census3 | 8 | 1 | 400 | 2 | 0 | 192 | 908 |
| chscase_census2 | 8 | 1 | 400 | 2 | 0 | 197 | 909 |
| plasma_retinol | 14 | 4 | 315 | 2 | 0 | 133 | 915 |
| visualizing_galaxy | 5 | 1 | 323 | 2 | 0 | 148 | 925 |
| colleges_usnews | 34 | 2 | 1302 | 2 | 7830 | 614 | 930 |

Table 9: Meta-Datasets used for developing the prior (continued).

| Name | #Feat. | #Cat. | #Inst. | #Class. | #NaNs | Minor. Class Size | OpenML ID |
|---|---|---|---|---|---|---|---|
| disclosure_z | 4 | 1 | 662 | 2 | 0 | 314 | 931 |
| socmob | 6 | 5 | 1156 | 2 | 0 | 256 | 934 |
| chscase_whale | 9 | 1 | 228 | 2 | 20 | 111 | 939 |
| water-treatment | 37 | 16 | 527 | 2 | 542 | 80 | 940 |
| lowbwt | 10 | 8 | 189 | 2 | 0 | 90 | 941 |
| arsenic-female-bladder | 5 | 2 | 559 | 2 | 0 | 80 | 949 |
| analcatdata_halloffame | 17 | 2 | 1340 | 2 | 20 | 125 | 966 |
| analcatdata_birthday | 4 | 3 | 365 | 2 | 30 | 53 | 968 |
| analcatdata_draft | 5 | 3 | 366 | 2 | 1 | 32 | 984 |
| collins | 23 | 3 | 500 | 2 | 0 | 80 | 987 |
| prnn_fglass | 10 | 1 | 214 | 2 | 0 | 76 | 996 |
| jEdit_4.2_4.3 | 9 | 1 | 369 | 2 | 0 | 165 | 1048 |
| mc2 | 40 | 1 | 161 | 2 | 0 | 52 | 1054 |
| mw1 | 38 | 1 | 403 | 2 | 0 | 31 | 1071 |
| jEdit_4.0_4.2 | 9 | 1 | 274 | 2 | 0 | 134 | 1073 |
| PopularKids | 11 | 5 | 478 | 3 | 0 | 90 | 1100 |
| teachingAssistant | 7 | 5 | 151 | 3 | 0 | 49 | 1115 |
| lungcancer_GSE31210 | 24 | 3 | 226 | 2 | 0 | 35 | 1412 |
| MegaWatt1 | 38 | 1 | 253 | 2 | 0 | 27 | 1442 |
| PizzaCutter1 | 38 | 1 | 661 | 2 | 0 | 52 | 1443 |
| PizzaCutter3 | 38 | 1 | 1043 | 2 | 0 | 127 | 1444 |
| CostaMadre1 | 38 | 1 | 296 | 2 | 0 | 38 | 1446 |
| CastMetal1 | 38 | 1 | 327 | 2 | 0 | 42 | 1447 |
| KnuggetChase3 | 40 | 1 | 194 | 2 | 0 | 36 | 1448 |
| PieChart1 | 38 | 1 | 705 | 2 | 0 | 61 | 1451 |
| PieChart3 | 38 | 1 | 1077 | 2 | 0 | 134 | 1453 |
| parkinsons | 23 | 1 | 195 | 2 | 0 | 48 | 1488 |
| planning-relax | 13 | 1 | 182 | 2 | 0 | 52 | 1490 |
| qualitative-bankruptcy | 7 | 7 | 250 | 2 | 0 | 107 | 1495 |
| sa-heart | 10 | 2 | 462 | 2 | 0 | 160 | 1498 |
| seeds | 8 | 1 | 210 | 3 | 0 | 70 | 1499 |
| thoracic-surgery | 17 | 14 | 470 | 2 | 0 | 70 | 1506 |
| user-knowledge | 6 | 1 | 403 | 5 | 0 | 24 | 1508 |
| wholesale-customers | 9 | 2 | 440 | 2 | 0 | 142 | 1511 |
| heart-long-beach | 14 | 1 | 200 | 5 | 0 | 10 | 1512 |
| robot-failures-lp5 | 91 | 1 | 164 | 5 | 0 | 21 | 1520 |
| vertebra-column | 7 | 1 | 310 | 3 | 0 | 60 | 1523 |
| Smartphone-Based... | 68 | 2 | 180 | 6 | 0 | 30 | 4153 |
| breast-cancer-... | 10 | 10 | 277 | 2 | 0 | 81 | 23499 |
| LED-display-... | 8 | 1 | 500 | 10 | 0 | 37 | 40496 |
| GAMETES_Epistasis... | 21 | 21 | 1600 | 2 | 0 | 800 | 40646 |
| calendarDOW | 33 | 21 | 399 | 5 | 0 | 44 | 40663 |
| corral | 7 | 7 | 160 | 2 | 0 | 70 | 40669 |
| mofn-3-7-10 | 11 | 11 | 1324 | 2 | 0 | 292 | 40680 |
| thyroid-new | 6 | 1 | 215 | 3 | 0 | 30 | 40682 |
| solar-flare | 13 | 13 | 315 | 5 | 0 | 21 | 40686 |
| threeOf9 | 10 | 10 | 512 | 2 | 0 | 238 | 40690 |
| xd6 | 10 | 10 | 973 | 2 | 0 | 322 | 40693 |
| tokyo1 | 45 | 3 | 959 | 2 | 0 | 346 | 40705 |
| parity5_plus_5 | 11 | 11 | 1124 | 2 | 0 | 557 | 40706 |
| cleve | 14 | 9 | 303 | 2 | 0 | 138 | 40710 |
| cleveland-nominal | 8 | 8 | 303 | 5 | 0 | 13 | 40711 |
| Australian | 15 | 9 | 690 | 2 | 0 | 307 | 40981 |
| DiabeticMellitus | 98 | 1 | 281 | 2 | 2 | 99 | 41430 |
| conference_attendance | 7 | 7 | 246 | 2 | 0 | 31 | 41538 |
| CPMP-2015-... | 23 | 1 | 527 | 4 | 0 | 78 | 41919 |
| TuningSVMs | 81 | 1 | 156 | 2 | 0 | 54 | 41976 |
| regime_alimentaire | 20 | 17 | 202 | 2 | 17 | 41 | 42172 |
| iris-example | 5 | 1 | 150 | 3 | 0 | 50 | 42261 |
| Touch2 | 11 | 1 | 265 | 8 | 0 | 27 | 42544 |
| penguins | 7 | 3 | 344 | 3 | 18 | 68 | 42585 |
| titanic | 8 | 5 | 891 | 2 | 689 | 342 | 42638 |

Table 10: Datasets used for the evaluation in the OpenML-AutoML Benchmark. These include all datasets with at most 1 111 samples, 100 features and 10 classes.

| Name | #Feat. | #Cat. | #Inst. | Class Size | #NaNs | Minor. Class Size | OpenML ID |
|---|---|---|---|---|---|---|---|
| credit-g | 21 | 14 | 1000 | 2 | 0 | 300 | 31 |
| vehicle | 19 | 1 | 846 | 4 | 0 | 199 | 54 |
| wine | 14 | 1 | 178 | 2 | 0 | 71 | 973 |
| blood-transfusion-service-center | 5 | 1 | 748 | 2 | 0 | 178 | 1464 |
| Australian | 15 | 9 | 690 | 2 | 0 | 307 | 40981 |

Table 11: Evaluation datasets for model generalization experiments.

| Name | #Feat. | #Cat. | #Inst. | #Class. | #NaNs | Minor. Class Size | OpenML ID |
|---|---|---|---|---|---|---|---|
| KDDCup09_appetency | 231 | 39 | 50000 | 2 | 8024152 | 890 | 1111 |
| airlines | 8 | 5 | 539383 | 2 | 0 | 240264 | 1169 |
| bank-marketing | 17 | 10 | 45211 | 2 | 0 | 5289 | 1461 |
| nomao | 119 | 30 | 34465 | 2 | 0 | 9844 | 1486 |
| adult | 15 | 9 | 48842 | 2 | 6465 | 11687 | 1590 |
| covertype | 55 | 45 | 581012 | 7 | 0 | 2747 | 1596 |
| numerai28.6 | 22 | 1 | 96320 | 2 | 0 | 47662 | 23517 |
| connect-4 | 43 | 43 | 67557 | 3 | 0 | 6449 | 40668 |
| jungle_chess_2pcs. | 7 | 1 | 44819 | 3 | 0 | 4335 | 41027 |
| APSFailure | 171 | 1 | 76000 | 2 | 1078695 | 1375 | 41138 |
| albert | 79 | 53 | 425240 | 2 | 2734000 | 212620 | 41147 |
| MiniBooNE | 51 | 1 | 130064 | 2 | 0 | 36499 | 41150 |
| guillermo | 4297 | 1 | 20000 | 2 | 0 | 8003 | 41159 |
| riccardo | 4297 | 1 | 20000 | 2 | 0 | 5000 | 41161 |
| volkert | 181 | 1 | 58310 | 10 | 0 | 1361 | 41166 |
| dionis | 61 | 1 | 416188 | 355 | 0 | 878 | 41167 |
| jannis | 55 | 1 | 83733 | 4 | 0 | 1687 | 41168 |
| helena | 28 | 1 | 65196 | 100 | 0 | 111 | 41169 |

