# OpenReview forum: "TabPFN: A Transformer That Solves Small Tabular Classification Problems in a Second"
_ICLR.cc/2023/Conference — ICLR 2023 notable top 25%_

### Official Review · Reviewer_gGgo · 2022-10-16

**Confidence:** 3
**Clarity, Quality, Novelty And Reproducibility:** The work is relatively clear and well…
**Correctness:** 3
**Technical Novelty And Significance:** 2
**Empirical Novelty And Significance:** 3
**Recommendation:** 6

**Strength And Weaknesses:**

Strengths
---
TabPFN takes a significantly different approach to tabular classification problems than many current deep-learning approaches, leveraging the transformer architecture and a novel data-generating prior to produce competitive predictions for new unseen tabular classification datasets in a fraction of a second.

Comparisons on toy datasets demonstrate TabPFN's ability to produce appropriate decision boundaries that sometimes better reflect the underlying dataset distribution than more traditional machine-learning approaches.

Experiments are performed on a large number (30) of tabular classification datasets.

Predictions for a new unseen test set require less than a second (using GPU).

Weaknesses
---
TabPFN is only suitable for small-scale tabular classification tasks (e.g., $\le$ 2000 examples), thus limiting TabPFN to larger problems.

The predictive performance of TabPFN and AutoGluon is similar (Table 1), and the combination of TabPFN (with ensembling) and AutoGluon tends to work best. However, TabPFN + AutoGluon seems only marginally better than AutoGluon on it own, questioning the necessity of adding a complex method such as TabPFN.

The empirical runtime comparisons between TabPFN and the competing methods is also not clearly described. Are the times in Table 1 showing the average total train+test time for the non-TabPFN methods? Also, are the non-PFN methods evaluated using a CPU or GPU? I think it would be useful to see training and test times as separate entities in Table 1, making it more clear where the tradeoffs exist between using TabPFN or a more traditional method like GBDTs or AutoGluon.

I think there could be a clearer discussion more precisely describing the differences between this work and the work by Muller et al. (2022) since this work is heavily inspired by that work. For example, the authors introduce their novel prior in Section 3 describing their use of SCMs and BNNs, but then also describe their use of BNNs follows the work by Muller et al. in Section 3.4. I think it also important to see a comparison to the work by Muller et al. in the experimental evaluation, or at least a discussion about why this comparison is not relevant.

Additional Comments
---
Are there any model size comparison results between TabPFN and competing methods? This information could further help ML practitioners when deciding whether to use TabPFN or a different method for their particular problem.

Minor Weaknesses
---
In Figure 1, the first description in the caption (Left (a)) is describing the right part of the figure, while the second description (Right (b)) is describing the left part of the figure.

Figures 3, 4, and 5 are not color-blind friendly.

Footnote 4 should go after the period.

**Summary Of The Paper:**

The authors propose TabPFN, a transformer-based prior-data fitted network trained to directly approximate the posterior predictive distribution (PPD), i.e., an infinitely large set of data-generating mechanisms. The main contribution of this work is the proposed prior, which is based on structural causal models (SCMs) and Bayesian neural networks (BNNs) in which distributions instead of point estimates are used for the hyperparameters of the prior. TabPFN is trained on 512 synthetic datasets generated from the prior and, once trained, can approximate the PPD for the proposed prior in a single forward pass for a set of target instances given a set of training examples. The result is a transfromer-based model trained to solve synthetically generated classification tasks from a tabular dataset prior in which predictions are very efficient.

Experiments on 30 small-scale classification datasets demonstrate TabPFN's ability to outperform gradient-boosted decision trees (GBDTs) and perform competitively with state-of-the-art AutoML approaches such as AutoGluon in terms of predictive performance while being significantly more efficient.

**Summary Of The Review:**

The work takes an interesting and relatively new approach towards small-scale tabular classification problems; with some refining, I think this paper can potentially make a significant contribution to the community.

---

> ### Author Response · Authors · 2022-11-15
> **Reply to Reviewer gGgo (Part 1)**
>
> Dear Reviewer gGgo,
>
> Thank you for your review and for acknowledging the significantly different approach we take toward tabular classification. We hope this will be an important step to improving classification on small tabular data! In the following, we will answer all your critiques one-by-one:
>
> You write:
> > “TabPFN is only suitable for small-scale tabular classification tasks”
>
> At first glance, tackling small datasets seems like a limitation, and we unfortunately have indeed communicated it this way. However, a large part of datasets, e.g. in medical applications, biological applications, medium-sized companies or anywhere where data is expensive to generate, tends to be small. We believe this "long-tail of datascience datasets" is very important to tackle and has been neglected. AutoML methods without hyperparameter tuning are especially important for these small datasets, as they are often tackled by non-expert users.
>
> Also, as you write, TabPFN is significantly different compared to established methods. Thus, it is not an approach that is tuned and tweaked to the maximum of its capacity. We expect PFNs to be scaled up to at least 10,000 examples in future work.
>
> You write:
> > "The predictive performance of TabPFN and AutoGluon is similar (Table 1), and the combination of TabPFN (with ensembling) and AutoGluon tends to work best. However, TabPFN + AutoGluon seems only marginally better than AutoGluon on it own, questioning the necessity of adding a complex method such as TabPFN."
>
> Here, we would like to strongly disagree with your analysis: While the change in accuracy is small on an absolute scale (+0.004), the changes in accuracy even from logistic regression to GBDT (logistic regression to LightGBM: +0.003) or from GBDT to AutoML methods (XGBosst to Auto-Sklearn: +0.002) is on that same scale. The impact of GBDT methods (e.g. XGBoost: 22k citations on google scholar) or AutoML methods is without a doubt very large. Also, we consider the average ranks, where improvements are much clearer: TabPFN + AutoGluon has an average rank 2.46, while AutoGluon by itself has a much worse average rank of 3.81.
> Furthermore, TabPFN takes only a fraction of the time compared to other methods to achieve similar performance (3127s/24.57s=127x faster than AutoGluon in Table 1 when TabPFN uses CPUs, and 3127s/0.4197s=7450x faster when TabPFN uses GPUs)!
>
> Importantly, TabPFN is a novel method, compared to the established methods that we compare to, which were developed and tuned over years (e.g. XGB has 560 contributors on GitHub and is sponsored by Nvidia and Intel, Autogluon and Autosklearn again are based on multiple of these base classifiers). We believe that we have shown an important step in creating a novel class of tabular classification algorithms and are very optimistic that TabPFN will improve further. Also, we are surprised that you call TabPFN a “complex method” compared to AutoGluon; modern AutoML methods, such as AutoGluon or Auto-sklearn, require code for many base classifiers, complex manual tuning and ensembling methods and, as a result are hard to maintain (with 93 contributors of AutoGluon and 79 contributors of Auto-sklearn) and have a host of code dependencies; in contrast, TabPFN is a single fixed transformer in which for a new dataset we only perform a single forward pass (plus an optional, unweighted ensembling step). It is hard to beat this in terms of code simplicity and portability.
>
> You write:
> > “The empirical runtime comparisons between TabPFN and the competing methods is also not clearly described.”
>
> This is a great point and we made sure to improve this in the manuscript in the meantime. All times described in the tables were for fitting and predicting combined; we have now clarified this in our main results: Table 1 and Figure 5.
> All baseline methods in the paper only had CPU cores available, while we evaluated both variants of the TabPFN with and without GPU and reported these times separately. We have a detailed discussion of the time comparisons in the “Results” paragraph of Section 5.2 and of our hardware setup in Appendix F.1. Additionally we added XGB tuned on GPU to our results in Figure 13 and 14.

---

> > ### Author Response · Authors · 2022-11-15
> > **Reply to Reviewer gGgo (Part 2)**
> >
> > You write:
> > > “I think there could be a clearer discussion more precisely describing the differences between this work and the work by Muller et al. (2022) since this work is heavily inspired by that work. For example, the authors introduce their novel prior in Section 3 describing their use of SCMs and BNNs, but then also describe their use of BNNs follows the work by Muller et al.”.
> >
> > We agree that we should more precisely discuss the methodological improvements compared to prior work. We were seeing things from our perspective, where we have been making constant improvements to that prior work and the many changes seemed obvious to us, and as a result neglected to emphasize this in our work. We have added Appendix C.2.6 detailing our changes and referenced it in Section 2 (Background) of the main paper. We have compiled a list of the most important contributions of our work here as well:
> >
> > - As you write, the SCM prior is completely new. And it was an important improvement to get good results, as we show in Table 4 of the Appendix. It pushes performance by 2% in this smaller scale setup, which is a bigger difference than between the final TabPFN and all baselines besides KNN and SAINT. We agree that this ablation result is very important and also agree that we should have emphasized it more in the paper. Thus we added a reference to it in Section 3.4 (BNN Prior).
> > - We evaluated pre-processing techniques for the TabPFN, which was not done at all before. We ensemble over different pre-processing pipelines, which include power transforms,outlier removal, mean/variance normalization and rotations of feature/class indices, e.g. `X = X[range(len(X))+4 % len(X)]`.
> > - As you write, we changed the model to make it faster. We shrank attention matrix sizes from `(n+m)^2` to `n^2 + n*m`, for `n` training points and `m` inference points
> > - We simulate categorical data (to some simple degree) to improve performance on datasets with such features.
> > - The PFNs for tabular data described in Muller et al. can only handle balanced binary datasets, whereas in Section 3.5. we show how to extend that prior to handle unbalanced data. Prior work was limited to 30 training samples, balanced binary-classification and 60 features. We scaled this up to 1000 training samples, 100 features, and 10 imbalanced classes, allowing it to directly tackle various datasets in practical applications, e.g., from the biomedical sector.
> > - Finally we detailed evaluations on different relevant benchmarks (OpenML CC-18 (Figure 5), OpenML-AutoML Benchmark (Appendix B.2)), analysis of model strengths and weaknesses (decision bounds (Figure 4), type of data (Appendix B.1), handling of SCM complexity (Appendix B.3.1) and robustness to random rotations/unnecessary features/dropping features (Appendix B.3.2/B.3.3), generalization to larger training set sizes (Appendix F.4)
> >
> > You write:
> > > "in Section 3.4. I think it also important to see a comparison to the work by Muller et al. in the experimental evaluation, or at least a discussion about why this comparison is not relevant."
> >
> > We mention the comparison even much earlier now in Section 2, paragraph “Tabular Data”.
> > We added this to Section 3.4.: "Müller et al. (2022) considered balanced binary datasets with at most 30 training samples and can thus not be applied to the datasets evaluated in our work.". An ablation of the adapted BNN prior (adapted for imbalanced multiclass and using the improvements detailed in Appendix C) is compared to the TabPFN prior in Table 4 in the Appendix. As mentioned above, it pushes performance by 2% in this smaller scale setup, which is a bigger difference than between the final TabPFN and all baselines besides KNN and SAINT.
> >
> > You write:
> > > “Are there any model size comparison results between TabPFN and competing methods? This information could further help ML practitioners when deciding whether to use TabPFN or a different method for their particular problem.”
> >
> > We do not have model size comparisons as it is hard to measure the size of TabPFN and baselines, including their code dependencies. However, the weights of TabPFN require 100MB of memory. This is not very big for most modern workstations, but might be a problem for edge devices. Does this answer your question or did we misunderstand?
> >
> > You write:
> > > "In Figure 1, the first description in the caption (Left (a)) is describing the right part of the figure, while the second description (Right (b)) is describing the left part of the figure.
> >
> > Figures 3, 4, and 5 are not color-blind friendly. Footnote 4 should go after the period."
> > Thank you for pointing us to this issue, we adapted the figures! We adapted coloring to be color-blind friendly (except for Figure 4, which uses red and blue for the plotting and should be color-blind friendly already?) and also introduced differing marker shapes for plotting scatterplots. We will additionally update figures in our Appendix for the camera-ready version.

---

> > > ### Author Response · Authors · 2022-11-15
> > > **Reply to Reviewer gGgo (Part 3)**
> > >
> > > We hope to have addressed your concerns and would gladly address additional questions or concerns you may still have. If we did address all your concerns, as we clarified several misunderstandings, we would appreciate a reflection of that in your score.
> > >
> > > For a diff of our revision, please see https://openreview.net/revisions/compare?id=cp5PvcI6w8_&left=utK4ygEOQo&right=dgAbsiFcj-&pdf=true

---

> > > > ### Comment · Reviewer_gGgo · 2022-11-16
> > > > **Response**
> > > >
> > > > I thank the authors for their response and for clarifying the majority of my concerns, and I have updated my score accordingly. However, I still think the clarity of the runtime comparison could be improved, by separating the time it takes to train vs. the time it takes to predict for TabPFN and each baseline model. Also, is the time required for pre-fitting TabPFN (20 hours) taken into account in Table 1? If not, it may be worth noting somewhere in those results.

---

> > > > > ### Author Response · Authors · 2022-11-17
> > > > > **Thank you for your quick response!**
> > > > >
> > > > > We try to resolve your last two open points below:
> > > > >
> > > > > *Splitting train/predict time*: TabPFN does training and prediction in the same, single forward-pass. The Transformer architecture used for TabPFN accepts both train and test data at the same time and thus these times cannot be reported separately. By modifying our architecture, it would be possible to untangle these steps, this is, however, out of scope for this work. We added this in our Limitations (Appendix A), as it might be relevant to split these calls for some users. We also separated the time it takes to train vs. the time it takes to predict, in Table 2 and added a reference to these times in Table 1.
> > > > >
> > > > > *Pre-training times*: We do not add the pre-training time of TabPFN to the table. This is fair, though.
> > > > >
> > > > > First, this is a one-time cost, i.e. only has to be done once. Our prior-fitting phase (20 x 8 GPU hours) is part of our algorithm development (i.e. developing ideas, writing code, evaluation) and users will not have to redo these steps. This might emphasize our point: For the rebuttal, we added evaluations on a different set of datasets (Figures 13,14,15 in the Appendix) and did not have to invest this cost again but reused the pre-trained model.
> > > > >
> > > > > Second, other methods have similar one-off upfront costs which we neither included in our tables.
> > > > > Two examples from the set of our baselines.
> > > > > First, building Auto-Sklearn for example involved figuring out what pipelines work well for what kind of datasets. This cost involved running hyper-parameter search for 24 hours on 140 datasets (= 3360 CPU hours) [1]. Our baseline, Auto-Sklearn 2.0, directly builds on this data [2].
> > > > > Second, XGBoost was initially released 2014 and the hyperparameter configurations that we used for it were found over years, by calibrating them on a large number of datasets. Also, the XGBoost algorithm itself has been refined over years. In that way, all methods have considerable one-time cost to yield the final classifier, which are then amortized over a large number of datasets.
> > > > >
> > > > > We appreciate your feedback very much and believe a detailed discussion of this point would also benefit our readers. We added a clarification to our results paragraph in Section 5.2. and added Appendix F.5 (Details on Time Comparisons), where we discuss both points.
> > > > >
> > > > > We hope this resolves even your last two points of criticism. If it does, we would appreciate it reflected in your score.
> > > > >
> > > > > [1] Auto-Sklearn 1.0 Feurer et al. (https://link.springer.com/chapter/10.1007/978-3-030-05318-5_6)
> > > > > [2] Auto-Sklearn 2.0 Feurer et al. (https://arxiv.org/abs/2007.04074)

---

### Official Review · Reviewer_pf2n · 2022-10-17

**Confidence:** 3
**Correctness:** 3
**Technical Novelty And Significance:** 3
**Empirical Novelty And Significance:** 4
**Recommendation:** 8

**Clarity, Quality, Novelty And Reproducibility:**

The presented approach seems novel, the authors provided the code for reproducibility and the datasets are public. However, it is difficult to verify the quality, as mentioned above,  since the pre-trained model requires providing the training and test data together, and it is difficult to verify that the test data is not used. It is also unclear if comparing to other models with an hour of hyperparameter search in a large space is better than just taking their default hyperparameters. The paper is generally clear, but the generation of the simulated data for obtaining the pretrained transformer should be explained in more detail, to assure no leakage from the datasets later used for evaluating the pre-trained model.

**Strength And Weaknesses:**

The paper presents an interesting new approach for the important problem of tabular data classification. The presented technique is limited to small datasets, as mentioned above, due to compute constraints (quadratic in size), but these smaller sizes could still be of interest. It is generally well-written.
A key weakness is that the pre-trained model requires providing the training and test data together. Even though the test data is not supposed to be used, it is difficult to verify this. Also, the generation of simulated data for the transformer training is not explained well, and it is important to understand how this was done, to make sure there was no leakage from the test datasets.
Another experiment that should have been done is using the default hyperparameters of Catboost, XGBoost and LightGBM, which may provide better results than an hour of search in a large hyperparameter space.



**Summary Of The Paper:**

The paper considers classification problems with small tabular datasets (a few thousand samples, and up to ten classes and 100 features). It presents a pre-trained model that solves such problems within a second, outperforming tree-ensembles and AutoML methods that run for an hour. The pre-trained model is a 12-layer transformer-based prior-data fitted network (PFN, introduced in Muller 2022), which uses a prior based on structural causal models. The pre-training uses 18000 batches of 512 simulated datasets. The training process takes 20 hours on an 8-GPU machine. Then the network is applied to 30 OpenML datasets (which takes only 1 sec on a GPU) and provides better ROCAUC results compared to XGBoost, Catboost, Auto-SkLearn and Auto-Gluon, which have a full hour for hyperparameter search. Some of these datasets had known state-of-the-art results (OpenML-AutoML) that were compared for validation.


**Summary Of The Review:**

This paper presents interesting results for an important problem. To accept it, I would like to see a code that does not require providing the training and test data together, and still obtains the same results. Another experiment that should be done is using the default Catboost and XGBoost hyperparameters, which may provide better results than an hour of search in a large hyperparameter space. Also, the simulated data for the transformer pre-training should be explained in more detail, to assure there was no leakage from datasets later used for the evaluation.

---

> ### Author Response · Authors · 2022-11-09
> **Reply to Reviewer pf2n (Part 1)**
>
> Dear Reviewer pf2n,
>
> Thank you very much for your review and the valuable contribution you are making to our work. We will answer your review first, as it seems most urgent to us. We will answer the other reviewers when we finish the experiments for each reviewer.
>
> **You write:**
> > “A key weakness is that the pre-trained model requires providing the training and test data together. Even though the test data is not supposed to be used, it is difficult to verify this.“
>
> There are two ways to interpret this, so we will try to answer both:
>
> i) You are concerned that the model uses label information from the test set. Our API follows the standard Scikit-learn interface (clf.fit(X_train, y_train); clf.predict(X_test)), which does not accept the test set labels. See our notebook for a minimal usage example, where you can check quickly and interactively that y_test is not passed: https://colab.research.google.com/drive/1J0l1AtMV_H1KQ7IRbgJje5hMhKHczH7-?usp=sharing
> We use this very interface to generate the TabPFN results (on the subset of OpenML CC-18 described in the paper) in the table below.
>
> ii) You are concerned that statistics of the test set inputs are used to improve classification, e.g. that the transformer also normalizes with respect to other test set inputs. We understand that this is hard to verify from looking at the repository – however, we made sure that this does not happen and also previously evaluated this.
> To make this easy to verify for you, we provide you with the following two ways.
> First, we show results for the following experiment: We trained and predicted using only a single test sample at a time. Thus the classifier cannot access any other test samples when making predictions. We checked performance per dataset and several other metrics and got the same results on the subset of OpenML CC-18 that we used in the paper:
>
> |                                 | TabPFN one-by-one   | TabPFN|
> |:--------------------------------|:----------------------------------|:-----------------------|
> | M. rank AUC OVO                 | 1.5                               | 1.5                    |
> | Mean rank Acc.                  | 1.5                               | 1.5                    |
> | Mean rank CE                    | 1.5                               | 1.5                    |
> | Mean ROC AUC                    | .894+-.01                         | .894+-.01              |
> | Mean Acc.                       | .826+-.012                        | .826+-.012             |
> | Mean Cross Entropy              | .732+-.018                        | .732+-.018             |
> | Mean Brier loss                 | .225+-.013                        | .225+-.013             |
> | Mean Expected Calibration Error | .0422+-.012                       | .0422+-.012            |
> | Mean time (s)                   | 266.14186100959785                | 1.0105533440907795     |
>
> Secondly, we enable you to verify this outcome easily using a notebook, which allows you to compare predictions for an example dataset when feeding the whole test set to the model compared to a one-by-one feeding: https://colab.research.google.com/drive/1uEPbk3S4AaAQWmp8yzys7wPttxUDCpP3?usp=sharing. You can see that even the predicted probabilities are the same up to rounding errors (< 1e-5). The predicted labels are an exact match for all splits that we tried.
>
> To make these exact matches possible, we made the TabPFN deterministic, with the following very small change: https://github.com/tabpfn-anonym/TabPFNAnonym/commit/546959f6f40eab9c951fe29dbc10b8ef227c0f5f (before there was some randomness in its predictions, and thus there were small differences between runs).
>
> **You write:**
> > “Also, the generation of simulated data for the transformer training is not explained well, and it is important to understand how this was done, to make sure there was no leakage from the test datasets.“
>
> Regarding a possible leakage of the test datasets: All the datasets in our prior (which is used for training the transformer) are completely synthetic and generated with a few hundred lines of code (mostly in https://github.com/tabpfn-anonym/TabPFNAnonym/blob/main/priors/mlp.py), with hyperparameters described in the paper and in https://github.com/tabpfn-anonym/TabPFNAnonym/blob/main/scripts/model_configs.py. This code implements the prior described in Section 3, which is a combination of ideas and models (SCMs, BNNs) rather than specific data. We do not train the weights of the TabPFN on real-world data at all. Thus, there is no chance of a leakage from the test sets.
>
> Regarding the explanation of the TabPFN prior: We see your point. To make it more clear from a procedural point of view, we show pseudo code to generate a synthetic training dataset with the SCM-based prior in Appendix C.1. This is a novel way to train a classifier, thus we are very happy to include any explanation suggestions you have to make it simpler.

---

> > ### Author Response · Authors · 2022-11-09
> > **Reply to Reviewer pf2n (Part 2)**
> >
> > **You write:**
> > > “Another experiment that should have been done is using the default hyperparameters of Catboost, XGBoost and LightGBM, which may provide better results than an hour of search in a large hyperparameter space.”
> >
> > We gladly comply with this request. The resulting plot, also comparing further baselines to TabPFN is here: https://github.com/tabpfn-anonym/TabPFNResults/blob/main/PlotAllTestDatasetsAllDefaults.png
> >
> > As you can see, on the 30 datasets from the OpenML CC-18 Benchmark, TabPFN outperforms all baselines by a wide margin, even when defaults are used. We will also add these results to the paper.
> >
> > You can reload pre-computed results for this experiment, view per dataset metrics and rerun the evaluations in this Colab: https://colab.research.google.com/drive/1NXkMR143CwX9yDInEv7Oomqy2TpM9Clz?usp=sharing
> >
> > While we understand your skepticism about our very strong results, we hope that our open source code, set of Colabs and demos, and comprehensive analysis convinces you that our claims are true. If you agree we would appreciate it if you increase your score to accept our paper. (If you do not agree we would be glad to hear about what we can do to convince you.)

---

> > > ### Comment · Reviewer_pf2n · 2022-11-17
> > > **Follow-up question**
> > >
> > > Thank you for your answers and additional experiments. They address my questions well, and I think you also provided good answers to the concerns of other reviewers.
> > > I have an additional follow-up question. Could you please clarify all the differences in data pre-processing between XGBoost and TabPFN? Could XGBoost perform better if the exact same pre-processing was used? (the average AUC difference was just 0.3%)

---

> > > > ### Author Response · Authors · 2022-11-18
> > > > **Thank you very much for your feedback!**
> > > >
> > > > Thank you very much for your feedback, we are happy that we could address your questions well!
> > > >
> > > > We tried to use all methods (both TabPFN and all baselines) in the best possible setting. Thus, we followed best practices in how to use all baselines.
> > > >
> > > > In the case of XGBoost and TabPFN, this means we do the same preprocessing steps for both methods already.
> > > > Missing values (https://xgboost.readthedocs.io/en/stable/faq.html#how-to-deal-with-missing-values) and data normalization (https://github.com/dmlc/xgboost/issues/357) is handled by both methods internally, so we do not impute or normalize. Categorical variables are encoded using the sklearn LabelEncoder for both methods.
> > > >
> > > > We tested XGBoost with a OneHot encoding for categoricals as well (using additional meta-information from OpenML with a list of categorical features per dataset). Whether to use OneHot or LabelEncoding is debated for XGBoost since OneHot Encoding increases the number of features that go into it. This can reduce accuracy and increases the training time per model. In our evaluation of the small OpenML-CC18 Benchmark, we found LabelEncoding for XGBoost to be superior after one hour of tuning each method:
> > > >
> > > > |  | Tuned XGBoost with OneHot Encoder	| Tuned XGBoost with LabelEncoder
> > > > | --- | --- | --- |
> > > > | Mean ROC AUC	| .889+-.013	| .891+-.011	|
> > > > | Mean Acc.	| .819+-.013	| .821+-.013	|
> > > > | Mean time (s)	| 3626.919	| 3304.140	|
> > > >
> > > > Also, the evaluated AutoML methods, AutoSklearn and AutoGluon, use highly optimized preprocessing pipelines for their integrated classifiers (which include XGBoost). This pipeline selects strong preprocessing techniques for each dataset and classifier.
> > > > Thus, even more complex, per dataset preprocessing is implicitly tested for XGBoost and the best preprocessing was not defined by us, but by the creators of the respective AutoML library. For a list of preprocessing technqiues contained in AutoSklearn see: https://github.com/automl/auto-sklearn/tree/master/autosklearn/pipeline/components/feature_preprocessing, Autogluon: https://auto.gluon.ai/dev/tutorials/tabular_prediction/tabular-feature-engineering.html

---

> > > > > ### Comment · Reviewer_pf2n · 2022-11-20
> > > > > **Thank you for your clarifications**
> > > > >
> > > > > Thank you for addressing my additional question. I have updated my score based on all your responses. I think the paper presents an interesting approach with interesting experimental results, and it would therefore interest the conference attendees. It is always difficult to fully verify that extensive experimental comparisons were exactly apples-to-apples, and that there was no information leakage from datasets used in the development. However, I think the authors provided convincing evidence that the accuracy of their new technique is at least on par with previous methods for small numeric tabular datasets, while providing several potential advantages.

---

### Official Review · Reviewer_BJ77 · 2022-10-23

**Confidence:** 5
**Correctness:** 3
**Technical Novelty And Significance:** 4
**Empirical Novelty And Significance:** 4
**Recommendation:** 6

**Clarity, Quality, Novelty And Reproducibility:**

The manuscript is clear and well written (though the profusion of acronyms impedes readibility). It is interesting research bringing new elements to the community. And the reproducibility is strong given the material shared.


**Strength And Weaknesses:**

In my eyes, the strength of this contribution is its originality. It contributes a novel incredient to tabular deep-learning research, and has promising benchmarks.

However, I worry about the benchmarks and claims of "much better performance" than XGBoost, LigthGBM in the introduction. Many practitionners have reported that the claims of tabular deep learning papers do not match their experience (which I found true with my own experimentation on many occasions). I honestly do not know why there is this disconnect, but it is not good press for our community. Along this line, Grinstztajn et al provided a benchmark which they claim is more realistic than many used in the literature. It would have been interesting to include this benchmark.

Tabular data have a form of rotational invariance (individual features are important). Is this enforced in the BNN sampling (I can see how it would appear in the SCM sampling).

Multi-class prediction: how is the noise added? A cursory description almost suggests that the link is deterministic: one specific configuration of X leading to a given class

Section 3.5: typo: unqiue

For lightGBM and XGBoost, the native handling of missing values should be used, rather than imputation

Which models do AutoML systems choose (to be able to give conclusion that situation various families of models)? This information would be interesting to add to the manuscript.



**Summary Of The Paper:**

This submission present tabPFN, an approach for fast learning on small tabular datasets. The approach uses a transformer to featurize the dataset and can learn with a signle pass forward through the network, without any backpropagation, which constitutes the biggest benefit of the approach. The model is an instance of "Prior-Data Fitter Network", fitted in a meta-learning style on rich simulated data. These data are generated via either Bayesian Neural Networks or Structural Causal Models, tuned to generate data similar to tabular data, with up to 2000 samples, 100 features and 10 imbalanced classes per dataset.

The methods is benchmarked on 30 small datasets from the OpenML-CC18 suite, comparing to established baselines, including ligthGBM and XGBoost. Benchmarks show better performance of the contributed model without fine-tuning or hyper-parameter selection, and in particular with very little time budget. *Edit* after rebuttal period, the authors added more datasets.

**Summary Of The Review:**

An interesting idea with very strong claims. I am not sure if the empirical evidence is quite as strong as the claims, and the claims should be moderated.

---

> ### Author Response · Authors · 2022-11-15
> **Reply to Reviewer BJ77 (Part 1)**
>
> Dear Reviewer BJ77,
>
> Thank you very much for thoroughly reading through our work and appreciating the potential and originality of it! We hope this work will contribute to the progress on tabular data classification. We would like to address your concerns one by one:
>
> You write:
> > "[..] I worry about the benchmarks and claims of "much better performance" than XGBoost, LigthGBM in the introduction. Many practitioners have reported that the claims of tabular deep learning papers do not match their experience (which I found true with my own experimentation on many occasions)."
>
> This is a very important concern to us and an observation we have also made. We made sure to evaluate much more rigorously than previous work on deep learning for tabular data.
>
> We evaluate 30 datasets from a public benchmark, using the same preprocessing steps and a wide range of baselines for each dataset. That means we compare on a much larger benchmark (that we didn't hand-pick). This is vastly different from e.g. TabNet, which evaluates 9 datasets, not taken from a benchmark suite, with custom preprocessing for each.
> In any case, we do *not* claim that TabPFN works best on every single dataset; in contrast, in our ablation in Figure 6 of the Appendix, we demonstrate that on datasets with categorical data or with missing values, TabPFN does not perform as well as gradient boosting or AutoML tuned for one hour. We added an explicit mention of this to the main results (5.2.Results). Furthermore, we did not optimize the preprocessing of any method on a per-dataset basis and are making no claim as to whether this may improve the performance of some of the baselines.
>
> To put some of the reviewers’ worries to rest, we have now additionally evaluated TabPFN against extra baselines and also on the 149 validation datasets. The results on these datasets are qualitatively similar to our existing results, both for datasets with only numeric features, and for those that include categorical features. We have added these results to the Appendix in Figures 13 and 14 in addition to critical difference plots for multiple scenarios (test/validation set, numerical/non-numerical data, different baselines) in Figure 15. You can find the code to rerun evaluations and per datasets results at https://colab.research.google.com/drive/1yUGaAf3D7RSyO5Jc4PYXSVbtaUAozh6S?usp=sharing
>
> Making progress on tabular datasets is very important (e.g. most scientific datasets are tabular). Thus even though progress using deep learning based methods has not been as fast as hoped, we can not just stop working on this set of methods. One reason why there might have been a gap between practitioners' and papers' claims could also be that practitioners are used to preparing data for GBDT-based methods and are most experienced in tuning those. Talking to practitioners about our work, we realized that it is very important to communicate the settings in which TabPFN works best very clearly. We have updated the README file on the shared GitHub repository to reflect these learnings and hope this will help to make TabPFN more successful with practitioners.
>
> You write:
> > "Along this line, Grinstztajn et al provided a benchmark which they claim is more realistic than many used in the literature. It would have been interesting to include this benchmark."
>
> Grinzstjan et al. benchmark medium-sized (< 10k samples) and large-sized datasets (< 50k samples) while we focus on small-sized datasets (< 1k samples); none of their benchmarks has less than 2k samples. Also, they preprocess datasets in various ways. Most notably, they create balanced binary tasks from the original multiclass datasets, which is less realistic than our imbalanced multiclass setting.

---

> > ### Author Response · Authors · 2022-11-15
> > **Reply to Reviewer BJ77 (Part 2)**
> >
> > You write
> > > "Tabular data have a form of rotational invariance (individual features are important). Is this enforced in the BNN sampling (I can see how it would appear in the SCM sampling)."
> >
> > Thanks for asking such in-depth questions. There might be a typo in your question: Tabular data is usually not rotationally invariant, and individual features are important. (Gael Varoquaux’s tweet about their benchmark paper contained the same typo: https://twitter.com/GaelVaroquaux/status/1549422445198774272 https://twitter.com/GaelVaroquaux/status/1549426496053428224)
> >
> > You are right in that with a prior based on SCMs, the algorithms learn not to be rotationally invariant. For our BNN prior, some points make it not rotationally invariant, but to a lesser degree. It is not rotationally invariant because in each layer, including the first layer, some connections are dropped at random. In the extreme case, only one connection of the first layer would be left, surely making this not rotationally invariant. Of course, most of the time, this will not happen. This aligns with our analysis in Figure 8 (Appendix) on the left, where we observe that TabPFN is more affected by rotations than MLPs but less so than GBDT methods.
> >
> > You write:
> > > "Multi-class prediction: how is the noise added? A cursory description almost suggests that the link is deterministic: one specific configuration of X leading to a given class"
> >
> > The SCM prior’s non-determinism arises in two ways. For BNNs and SCMs, noise is added to all intermediate nodes, which makes the values observed in other nodes non-deterministic in all cases. Since the labels are sampled from one such node, it will be non-deterministic. For further reference, we refer to Equation 3. We added a reference to the SCM equation in the description of our prior, to hopefully make this clearer.
> >
> > You write
> > > "For lightGBM and XGBoost, the native handling of missing values should be used, rather than imputation"
> >
> > We already use their native handling of missing values. We write, "Where necessary, we imputed missing values with the mean, one-hot encoded categorical inputs, normalized features, and passed categorical feature indicators.", as you note, for LightGBM and XGBoost, this is *not* necessary.
> >
> > We double-checked our implementation, and that is exactly what we do already. See:
> > https://github.com/tabpfn-anonym/TabPFNAnonym/blob/main/scripts/tabular_baselines.py#L988 (LightGBM)
> > https://github.com/tabpfn-anonym/TabPFNAnonym/blob/main/scripts/tabular_baselines.py#L1358 (XGBoost)
> > Both times we set `impute=False`, which means that missing values will be retained.
> >
> > You write:
> > > "Which models do AutoML systems choose (to be able to give conclusion that situation various families of models)? This information would be interesting to add to the manuscript."
> >
> > We analyzed this for the OpenML CC-18 Benchmark for all splits, and you can view our results for autosklearn at: https://colab.research.google.com/drive/16TXbtY5JwIPA9ViAzOtjfcqvUv3eHdV2?authuser=1#scrollTo=08F2ZMYtyqYa
> > You can see that autosklearn uses and ensembles various classifiers (adaboost,libsvm_svc,bernoulli_nb,decision_tree,multinomial_nb,passive_aggressive,random_forest,k_nearest_neighbors,liblinear_svc,sgd,extra_trees,gaussian_nb,lda,mlp); we now added this analysis to our paper in Figure 12. We wish not to add too many figures in the manuscript and keep things as simple as possible, so we limited this analysis to autosklearn as a representative AutoML method. (Also, AutoGluon always uses the same handcrafted base-level methods and uses different multi-level stack ensembles, so it gets much harder to quantify which base model is most important for it.)
> >
> > We hope to have addressed your concerns and would gladly address additional questions or concerns you may still have. If we did address all your concerns, as we clarified several misunderstandings, we would appreciate a substantial reflection of that in your score.
> > For a diff of our revision, please see https://openreview.net/revisions/compare?id=cp5PvcI6w8_&left=utK4ygEOQo&right=dgAbsiFcj-&pdf=true

---

> > > ### Comment · Reviewer_BJ77 · 2022-11-23
> > > **A very interesting and promising method, with a slight oversell**
> > >
> > > I thank that reviewers for answering my concern. I really appreciate the added figures, which make the empirical results more convincing.
> > >
> > > Fig 14 shows that on data that include at least categorical feature, XGBoost gvies a better tradeoff time/performance.
> > >
> > > I do think that it is important to reflect this reality in a more nuanced position with regards to tree models. For instance, the abstract of the manuscript still states: "On 30 small datasets from the OpenML-CC18 suite, we show that our method clearly outperforms boosted trees". While this sentence is true, it is not a good summary of the full empirical results and does not convey the right message.
> > >
> > > To summarize my view on this contribution: it is a very interesting and promising idea. Currently it overstate the performance of the method compared to boosted tree. In this respect, the tone of the paper must be moderated. However, I am convince that it is a worthwhile scientific contribution and will raise my rating accordingly (trusting that the authors will adjust the final version of the manuscript).

---

> > > > ### Comment · Reviewer_pf2n · 2022-11-23
> > > > **I agree with you**
> > > >
> > > > The abstract should clearly state that the improved results compared to tree ensembles are for numeric data (not for categorical data nor for data with unknowns).

---

> > > > > ### Author Response · Authors · 2022-11-25
> > > > > **Thank you very much for your feedback**
> > > > >
> > > > > Thank you for your feedback! We hope to have answered your concerns in our reply to reviewer BJ77, please see above.

---

> > > > ### Author Response · Authors · 2022-11-25
> > > > **Thank you very much for your feedback!**
> > > >
> > > > Thank you for your feedback, we were also recently thinking along the same lines, i.e., focusing this work on numerical features, but were reluctant to make this larger change unprompted. To avoid a misunderstanding, please note that our initial claim holds true: aggregated across the 30 test datasets, TabPFN statistically significantly outperforms boosted trees (see critical difference diagrams: https://anon.to/Jdnqpm and https://anon.to/2z63je). These results are mainly carried by the strong performance on numerical features, for which our prior was designed, and this is where TabPFN really shines. After evaluating and comparing the aggregate of 179 datasets in this rebuttal (see Figures 14, 15, and 16), we have more solid statistical evidence on the subgroups in which TabPFN works best (i.e. numerical features). We agree that by separating these two groups, we make the limitations on categorical data clearer while observing larger improvements on numerical data. Thank you for this suggestion, which will definitely add to the clarity of our work and claims. A similar point concerns the existence of missing data (which we had also ablated in Figure 6), and to handle these limitations consistently, we would focus the paper on numerical data without missing values.
> > > >
> > > > Our proposal for the updated abstract would be the following (with changes bold-faced):
> > > >
> > > > We present TabPFN, a trained Transformer that can do supervised classification for small tabular datasets in less than a second, needs no hyperparameter tuning and is competitive with state-of-the-art classification methods. TabPFN is fully entailed in the weights of our network, which accepts training and test samples as a set-valued input and yields predictions for the entire test set in a single forward pass. TabPFN is a Prior-Data Fitted Network (PFN) and is trained offline once, to approximate Bayesian inference on synthetic datasets drawn from our prior. This prior incorporates ideas from causal reasoning: It entails a large space of structural causal models with a preference for simple structures. On **the 18 small datasets in** the OpenML-CC18 suite, **that contain up to 1000 training data points, up to 100 purely numerical features without missing values, and up to 10 classes**, we show that our method clearly outperforms boosted trees and performs on par with complex state-of-the-art AutoML systems with up to 70x speedup. This increases to a 3200x speedup when a GPU is available. **We also validate these results on an additional 67 small numerical datasets from OpenML.** We provide all our code, the trained TabPFN, an interactive browser demo and a Colab notebook at https://github.com/tabpfn-anonym/TabPFNAnonym.
> > > >
> > > > We adapted Figure 5, which compares all methods with increasing time budgets, to limit the comparison to numerical datasets and likewise adapt Table 1, leaving the aggregate plots across 30 test datasets (and 179 valid + test datasets) in our Appendix.  We have statements in our conclusion as well as limitations regarding categorical features. We would be happy to share this new version, which we have drafted already, via the anonymous repository (updates to the paper are disabled in OpenReview in the current phase), but would like to refrain from doing so without being prompted in case this would be seen as inappropriate during this phase of the rebuttal.
> > > >
> > > > Thank you once again for your helpful engagement!

---

### Official Review · Reviewer_zM87 · 2022-10-24

**Confidence:** 4
**Correctness:** 3
**Technical Novelty And Significance:** 2
**Empirical Novelty And Significance:** 2
**Recommendation:** 8

**Clarity, Quality, Novelty And Reproducibility:**

The paper is easy to read and technically sound. It presents a new approach with state-of-the-art performance for limited datasets. Furthermore, the authors provide code which helps on reproducibility.

**Strength And Weaknesses:**

The paper presents an approach based on PFNs, where the authors pre-train their network with synthetic data obtaining great results compared to other state-of-the-art approaches. They present a data generation approach that successfully transfers different hypothesis to the TabPFNs.

There are however claims such as "Predictions based on causal reasoning" that should be better substantiated. As the authors do not demonstrate this is indeed happening. Just because the synthetic dataset comes from an SCM, is no guarantee that TabPFN is doing causal reasoning.

**Summary Of The Paper:**

In this paper, the authors introduce TabPFN. A transformer-based neural network for classification. Unlike traditional supervised approaches, the network is pre-trained to run classification on unseen datasets.
This transformer approach, runs inference on input data to produce a classification result. However, this input data contains the training and prediction datasets removing the need to do pre-training for a given dataset-classification task.

TabPFN is pre-trained on different synthetic datasets for the task of learning to obtain a hypothesis and use the hypothesis to make predictions. The limitations of the approach are clear, the dimensionality of the datasets is constrained by the transformer architecture, and therefore the scalability during training and inference grows quadratically. Nevertheless, this "ready out-of-the-box approach" offers state-of-the-art performance when compared to other classifiers.

**Summary Of The Review:**

The authors present a synthetic generator as well as a change in the transformer mask of PFNs to create a strong tabular classificator. They evaluate against state-of-the-art approaches and demonstrate increased accuracy as well as a significant reduction in time.

In general, I find the approach very interesting. However, I'm trying to understand the significance of the contribution compared to the PFNs paper. Indeed, the pre-training using SCMs is a significant portion of the contribution. However training on BNNs was already presented before and the speed characteristics are coming from PFNs.

As you mention in Fig. 1 that plots are based on the PFN paper, it would be good to repeat the same for Fig. 2, as there is an overlap.

---

> ### Author Response · Authors · 2022-11-15
> **Reply to Reviewer zM87**
>
> Dear Reviewer zM87,
>
> Thank you very much for your thoughtful review and critical reading. We are pleased that you see the impact our work could have and that you find our paper to be clearly written. We hope we can adequately address your questions in the following:
>
> You write:
> > “However, I'm trying to understand the significance of the contribution compared to the PFNs paper.”
>
> We agree that we should more precisely discuss the methodological improvements compared to prior work. We were seeing things from our perspective, where we have been making constant improvements to that prior work and the many changes seemed obvious to us, and as a result neglected to emphasize this in our work. We have added Appendix C.2.6 detailing our changes and referenced it in Section 2 (Background) of the main paper. We have compiled a list of the most important contributions of our work here as well:
>
> - As you write, the SCM prior is completely new. And it was an important improvement to get good results, as we show in Table 4 of the Appendix. It pushes performance by 2% in this smaller scale setup, which is a bigger difference than between the final TabPFN and all baselines besides KNN and SAINT. We agree that this ablation result is very important and also agree that we should have emphasized it more in the paper. Thus we added a reference to it in Section 3.4 (BNN Prior).
> - We evaluated pre-processing techniques for the TabPFN, which was not done at all before. We ensemble over different pre-processing pipelines, which include power transforms,outlier removal, mean/variance normalization and rotations of feature/class indices, e.g. `X = X[range(len(X))+4 % len(X)]`.
> - As you write, we changed the model to make it faster. We shrank attention matrix sizes from `(n+m)^2` to `n^2 + n*m`, for `n` training points and `m` inference points
> - We simulate categorical data (to some simple degree) to improve performance on datasets with such features.
> - The PFNs for tabular data described in Muller et al. can only handle balanced binary datasets, whereas in Section 3.5. we show how to extend that prior to handle unbalanced data. Prior work was limited to 30 training samples, balanced binary-classification and 60 features. We scaled this up to 1000 training samples, 100 features, and 10 imbalanced classes, allowing it to directly tackle various datasets in practical applications, e.g., from the biomedical sector.
> - Finally we detailed evaluations on different relevant benchmarks (OpenML CC-18 (Figure 5), OpenML-AutoML Benchmark (Appendix B.2)), analysis of model strengths and weaknesses (decision bounds (Figure 4), type of data (Appendix B.1), handling of SCM complexity (Appendix B.3.1) and robustness to random rotations/unnecessary features/dropping features (Appendix B.3.2/B.3.3), generalization to larger training set sizes (Appendix F.4)
>
> You write:
> > “There are however claims such as ‘Predictions based on causal reasoning’ that should be better substantiated. As the authors do not demonstrate this is indeed happening.“
>
> This is a good point, thank you for bringing it up. We did not intend to imply that TabPFN is performing causal reasoning or causal inference. Admittedly this has been unclear in the formulation that you bring up. This formulation was made just once and otherwise we write "Predictions based on ideas from causal reasoning", which we now adopted everywhere. In Section 3.3, paragraph "Predictions based on ideas from causal reasoning", we detail how our work can be related to the causal reasoning literature. We explicitly do not claim that we do causal inference or discovery, nor causal reasoning. We updated this section slightly, to make it even clearer that we only train on artificial data, which is built by a process inspired by SCMs. Figure 7 in our appendix shows, that our predictions align with simple SCM hypotheses, which is the claim we intend. We also referenced this figure in Section 3.3 (SCM Prior). Does this adequately answer your question, and do you believe Section 3.3 (SCM Prior) now adequately reflects our claims?
>
> You write:
> > “As you mention in Fig. 1 that plots are based on the PFN paper, it would be good to repeat the same for Fig. 2, as there is an overlap.”
>
> Thanks, we added this.
>
> We hope to have addressed your concerns and would gladly address additional questions or concerns you may still have. If we did address all your concerns, as we clarified several misunderstandings, we would appreciate a reflection of that in your score.
>
> For a diff of our revision, please see https://openreview.net/revisions/compare?id=cp5PvcI6w8_&left=utK4ygEOQo&right=dgAbsiFcj-&pdf=true

---

> > ### Comment · Reviewer_zM87 · 2022-11-16
> > **Good notes**
> >
> > I appreciate your effort in clarifying and improving the paper, and I will reflect the scores accordingly.
> >
> > I've been thinking about your work, more precisely in the context of the limitations on the data sizes. Would it make sense to use your model in a sliding window fashion over a whole dataset?
> > Consider a larger dataset with 50k instances. You could iterate over the 50k instances, 1k instances at the time, making predictions for a given set of instances fixed.
> > This way, you end up with 50 different predictions for any given test instance. At this point you can then merge the predictions in a mixture model way.
> > Indeed, rare cases would fail, but under certain amount of regularity in the data, this could actually work. Especially, if you over sample like in bootstrapping.
> > Alternatively, you could pair the training of your model in a similar way for larger datasets.
> >
> > Would this be a good idea to break the current size limitation of your approach? Granted, it's not as fast, but since your method is already pretty performant, this could open new possibilities.

---

> > > ### Author Response · Authors · 2022-11-17
> > > **Thank you very much!**
> > >
> > > We have been working on something similar since this does indeed seem very promising! However, we used a simple soft average ensembling in the end and, so far, this did not yield better results than our baselines. As you say training data might not be so regular and so information needs to be integrated. Also, we did not consider any smarter ways to combine the predictions and soft averaging might neglect that some splits contain less relevant information. We hope there would be progress in this direction as well.

---

### Decision · Program_Chairs · 2023-01-20

**Decision:**

Accept: notable-top-25%

**Justification For Why Not Higher Score:**

The main reason is the scope. I personally find that tabular data is an under-researched area in deep learning, but the fact is that many of the people in ICLR will not find this paper relevant because of the topic.


**Justification For Why Not Lower Score:**

The paper provides a novel approach to a highly explored area of tabular classification. This is a tough area to obtain high quality results given the great performance of tree-ensemble methods. Given the novelty of the approach, I think the ICLR audience would be interested in this paper, and think it deserves a spotlight.

**Metareview: Summary, Strengths And Weaknesses:**

The paper provides a neural network for classification that is pre-trained on synthetic data to be applied on unseen datasets, quickly and without hyperparameter tuning. The reviews agree that the idea is novel in that it is quite different than existing approaches for tabular data classification. Additionally, the empirical results seem to be thorough and convincingly show that the method provides high quality results on small datasets. One main drawback of the method is its limitation to small datasets, but I agree with the authors claim that (1) even with this limitation, the scope of influence is large as small datasets are quite common, and (2) given the originality of the paper, the idea could be extended to handle larger datasets in the future.

In the reviews, a few more concerns were raised regarding the representation of the test datasets of real-world problems, the paper's clarity, and some details of the algorithms. These were addressed in the rebuttal phase and seem to have been resolved. Overall, the paper is novel and shows promising results. I believe the scope of influence is great and that it would be a great addition to ICLR. I do urge the authors to carefully go over the reviews and discussions with the reviewers and incorporate the needed change to a camera-ready version.



**Note From Pc:**

if the above contains the word "oral" or "spotlight" please see: "oral" presentation means -> notable-top-5% and "spotlight" means -> notable-top-25%. As stated in our emails, we are disassociating presentation type from AC recommendations